# Implementing spatially explicit wind-driven seed and pollen dispersal in the individual-based larch simulation model: LAVESI-WIND 1.0

Stefan Kruse[1], Alexander Gerdes[1,2], Nadja J. Kath[3], Ulrike Herzschuh[1,3,4]

[1]Polar Terrestrial Environmental Systems Research Group, Alfred Wegener Institute Helmholtz Centre for Polar and Marine Research, 14473 Potsdam, Germany

[2]Institute of Physics and Astronomy, University of Potsdam, 14476 Potsdam, Germany

[3]Institute of Biology and Biochemistry, University of Potsdam, 14476 Potsdam, Germany

[4]Institute of Earth and Environmental Science, University of Potsdam, 14476 Potsdam, Germany

*Correspondence to*: Stefan Kruse (stefan.kruse@awi.de)

**Abstract.** It is of major interest to estimate the feedback of arctic ecosystems to the global warming we expect in upcoming decades. The speed of this response is driven by the potential of species to migrate, tracking their climate optimum. For this, sessile plants have to produce and disperse seeds to newly available habitats, and pollination of ovules is needed for the seeds to be viable. These two processes are also the vectors that pass genetic information through a population. A restricted exchange among subpopulations might lead to a maladapted population due to diversity losses. Hence, a realistic implementation of these dispersal processes into a simulation model would allow an assessment of the importance of diversity for the migration of plant species in various environments worldwide. To date, dynamic global vegetation models have been optimised for a global application and overestimate the migration of biome shifts in currently warming temperatures. We hypothesise that this is caused by neglecting important fine-scale processes, which are necessary to estimate realistic vegetation trajectories. Recently, we built and parameterised a simulation model LAVESI for larches that dominate the latitudinal treelines in the northernmost areas of Siberia. In this study, we updated the vegetation model by including seed and pollen dispersal driven by wind speed and direction. The seed dispersal is modelled as a ballistic flight, and for the pollination of ovules of seeds produced, we implemented a wind-determined and distance-dependent probability distribution function using a von Mises distribution to select the pollen donor. A local sensitivity analysis of both processes supported the robustness of the model's results to the paramterisation, although it highlighted the importance of recruitment and seed dispersal traits for migration rates. This individual-based and spatially explicit implementation of both dispersal processes makes it easily feasible to inherit plant traits and genetic information to assess the impact of migration processes on the genetics. Finally, we suggest how the final model can be applied to substantially help in unveiling the important drivers of migration dynamics and, with this, guide the improvement of recent global vegetation models.

## 1. Introduction

How fast vegetation communities can follow their shifting climate envelope in a changing environment is determined by their ability to migrate. This is exceptionally challenging under current global change and plants might strongly lag behind their moving climate envelope (Harsch et al., 2009; Loarie et al., 2009; Moran and Clark, 2012). Temperatures are increasing most strongly in the Arctic. Accordingly, forests in the tundra-taiga transition zone are expected to respond by migration into the tundra (Bader, 2014; Holtmeier and Broll, 2005; MacDonald et al., 2008). However, empirical studies show diverse responses to the warming, including treelines being stable, advancing or even retreating (Harsch et al., 2009). A taiga range expansion though, might positively feedback to a global temperature increase due to albedo reduction (Bonan, 2008; Piao et al., 2007; Shuman et al., 2011).

To predict forest responses to climate, computer models were designed with different scopes of complexity, between highly general to very specific (Grimm and Railsback, 2005; Thuiller et al., 2008). Among these, simulation studies with dynamic global vegetation models (DGVMs) tend to overestimate the turnover of treeless tundra into forests (Brazhnik and Shugart, 2015, 2016; Frost and Epstein, 2014; Kaplan and New, 2006; Roberts and Hamann, 2016; Sitch et al., 2008; Snell, 2014; Yu et al., 2009; Zhang et al., 2013). On the other hand, forest landscape models (e.g. Snell et al., 2014; Shifley et al., 2017; Epstein et al., 2007) and small-scale models (forest-gap or individual-based) provide sufficient detail to realistically represent the responses at a stand level, but need much effort for parameterisation, have higher computational expenses, and are therefore typically not applied over large areas (Martínez et al., 2011; Pacala et al., 1996; Pacala and Deutschman, 1995; Zhang et al., 2011) or lack the implementation of wind-driven seed and pollen dispersal (e.g. Epstein et al., 2007). Further problems of DGVMs arise from the use of plant functional types as they consist of species with a wide variety of traits (e.g. Lee 2011, Snell et al. 2014, Svenning et al. 2014). Nonetheless, the ability to form a closed canopy forest depends mainly on species traits acting at a fine-scale level such as (1) time needed to mature (life-cycle, high generation time) and produce viable seeds, (2) dispersal distance and the chance for long-distance seed dispersal and (3) germination and establishment of new individuals (Svenning et al., 2014). One source of the overestimation of migration rates of DGVMs is the unconstrained seed availability when climate variables allow a vegetation type to establish, which was recently pointed out by using a dispersal function between the grid points in simulations with a DGVM (Snell, 2014; Snell and Cowling, 2015). However, connecting grid cells to allow dispersal among them increases the computational complexity of such models (e.g. Nabel 2015), but would be necessary to simulate realistic large-scale vegetation responses. In addition, the structure of a tree stand, and its response to changes in external forcing, is determined by further local processes, such as spatially explicit competition among individuals of all ages and their interactions. Of special interest is the adaptation of the traits of individuals of local populations, which are influenced by gene flow through seed or pollen distributions across populations. High exchange can lead to outcrossing that hinders local adaptation, but also prevents negative consequences from diversity losses caused by inbreeding within isolated populations due to founder effects in the process of colonisation over large distances (Austerlitz et al., 1997; Burczyk et al., 2004; Fayard et al., 2009; Nishimura and Setoguchi, 2011; Ray and Excoffier, 2010). These processes have, so far, not been implemented continuously over a large scale in simulation models.

During the past decades treeline stands in the Siberian Arctic were densifying, but only rather slowly colonising the tundra (Frost et al., 2014; Kharuk et al., 2006; Montesano et al., 2016), which could be attributed to seed limitation (Wieczorek et al., 2017). We developed the *Larix* **ve**getation **si**mulator LAVESI to simulate tree stand dynamics at the Siberian treeline on the southern Taymyr Peninsula and use it as a framework to explore impacts of climate change on larch forests (Kruse et al., 2016). In the first version, the dispersal function randomly dispersed seeds by a probability density function describing a Gaussian term with a fat-tail. This could be parameterised to fit observed stand patterns. The model simulates tree stands on plots, representing a homogeneous forest, which can easily be enlarged to simulate wider areas. However, for simulations on larger transects passing from forests to treeless areas, wind direction and strength become more important for seed dispersal and needed to be included in the model. Seed dispersal processes are well studied (Nathan et al., 2011a; Nathan and Muller-Landau, 2000) and are sometimes implemented in vegetation models but rarely coupled with wind speed and direction (e.g. Lee, 2011; Levin et al., 2003; Snell, 2014). Also wind patterns might change over time, as the pressure levels vary in a changing climate (Trenberth, 1990), or are directed (Lisitzin, 2012) so that an implementation of wind-dependent dispersal would enable a more realistic simulation of migration (cf. Nathan et al., 2011b).

The new spatially explicit pollination function  tracks the full genealogy of a simulated tree stand and furthermore allows the inheritance of individually varying traits of each tree, rather than randomly drawing the actual trait value from the pool of available traits (cf. Scheiter et al., 2013). Additionally, the implementation of spatially explicit seed dispersal and pollination would enable us to align the model to detailed biogeographical knowledge gained from molecular methods (e.g. Navascués et al., 2010; Polezhaeva et al., 2010; Semerikov et al., 2007, 2013; Sjögren et al., 2017). We started with a very detailed small-scale model that can later be used to inform large-scale models especially about plot connectivity through seed dispersal and pollination and subsequent gene flow in landscapes.

We aim with this study to enable the simulation of spatially explicit and wind-dependent seed dispersal and pollination in the individual-based model LAVESI. After the coupling and verification of the seed dispersal kernel to prevailing winds and the incorporation of the pollination we test the model's sensitivity to its parameterisation in local sensitivity analyses and the influence on stand development, migration rates, and pollination distances.

## 2.   Methods

### 2.1.   General model description of the *Larix* vegetation simulator LAVESI

LAVESI is an individual-based spatially explicit model that currently simulates the life cycle of larch species as completely as possible from seeds to mature trees (Kruse et al., 2016). It was set up to improve our understanding of past and future treeline displacements under changing climates, focusing on the open larch forest ecosystem in northern Siberia, which is underlain by permafrost. The relevant processes (growth, seed production and dispersal, establishment and mortality) are incorporated as submodules, which were parameterised on the basis of field evidence and complemented with data from

literature. Simulation runs proceed in yearly time steps and are forced by monthly temperature and precipitation time series. The area simulated represents spatially homogeneous forest plots of variable size with the use of an environment grid (e.g. competition) with 20-cm tiles and where the handling of seeds dispersed beyond the plot borders can be set to deletion or reintroduction from the other side to simulate a forest patch. The model is programmed in C++ using standard template libraries. This and its modular structure allow a straightforward implementation of further extensions.

The model was successfully applied to conduct temperature-forcing experiments, where simulations revealed that the responses of the larch tree stands in Siberia – densification and northwards migration – could lag the applied hypothetical warming by several decades, until the end of 21$^{st}$ century (Kruse et al., 2016; Wieczorek et al., 2017).

Here we present the implementation of wind-dependent seed dispersal as well as the newly introduced pollination. The absorbing boundary condition had to be revised to allow the simulation of larger areas. Hence, we introduce a new mode of periodic boundary conditions that allows seeds leaving the simulated area (100 x 100 m) to reenter on the opposite side, so that the borders of a simulation plot are connected along all borders. This mimics a tree stand within a homogeneous forest, similar to forest gap models (e.g. Brazhnik and Shugart, 2016; Pacala et al., 1996; Pacala and Deutschman, 1995; Zhang et al., 2011) and we used it in the simulations used for verification and paramterisation for this manuscript. A second mode was implemented for simulations of hypothetical north-south transects (100 x 1,000 m), which were used in the sensitivity analyses, allowing seed dispersal only on the meridional borders but not the latitudinal limits.

## 2.2. Implementing dispersal processes coupled to wind speed and direction

### 2.2.1. Pollination probability

Pollen was not represented in the former LAVESI version, but is needed to independently track gene flow by seeds and pollen through time. Accordingly, Figure 1 illustrates how we implemented an individual based pollination for each seed's ovule using a wind-determined and distance-dependent probability distribution function for pollen dispersal (similar to Gregory, 1961). It makes use of the von Mises distribution, which is an angular equivalent to the Gaussian normal distribution, for the two-dimensional representation (Abramowitz and Stegun, 2012).

A pollen dispersal function was newly implemented as a distance-dependent probability function for pollination of each individual seed's ovule, rather than simulating the large amount of pollen released by each tree (Gregory, 1961; Kuparinen et al., 2007). For each seed-bearing tree, the probability of pollen donating trees is calculated and out of the list of potential fathers for each seed one tree is randomly determined according to this probability. The pollination probability of each seed's ovule on a tree is proportional to the amount of pollen in the air column around it, which is, for simplification in the current implementation, not additionally dependent on the performance of the tree so that every tree that bears cones is taken into account. This aspect might be included in future versions. The following function is used here as the distance-dependent probability distribution of arriving pollen:

$$p_r = exp\left(\frac{-2p_e r^{1-0.5m}}{\sqrt{\pi}C(1-0.5m)}\right) \tag{1}$$

where $r$ is the distance in m, $p_e$ is the ratio of pollen descending velocity $V_{d,pollen}$ estimated for *Larix gmelinii* (Eisenhut, 1961) and wind speed $V_w$ and Gregory's parameters $C$ and $m$ are set to $C = 0.6$ cm$^{-(1-0.5\ m)}$ and $m = 1.25$ (Eq. page 167 in Gregory, 1961).

The probability distribution $p_r$ described in Eq. 1 is multiplied by the von Mises distribution (Eq. 2), a continuous probability distribution on the circle, to include pollen distribution over a certain area and couple the process to the wind direction (illustration in Fig. 1; Abramowitz and Stegun, 2012).

$$p_v = \frac{exp\left(\kappa cos(\theta - \overline{\theta})\right)}{2\pi I_0(\kappa)} \tag{2}$$

where $\kappa$ is the inverse of the von Mises distribution's variance, and $I_o(\kappa)$ is the modified Bessel function of order 0 as a function of $\kappa$, $\theta$ is the angle between trees and $\overline{\theta}$ the actual wind direction. The modified Bessel function in the von Mises distribution is programmed in its integral representation using the Simpson integration scheme (Abramowitz and Stegun, 2012).

Consequently, following Gregory (1961) the pollination probability of a seed's ovule is:

$$p = p_r p_v = exp\left(\frac{-2p_e r^{1-0.5m}}{\sqrt{\pi}C(1-0.5m)}\right)\frac{exp\left(\kappa cos(\theta - \overline{\theta})\right)}{2\pi I_0(\kappa)}. \tag{3}$$

### 2.2.2. Seed dispersal

In the initial version of LAVESI, seeds are dispersed in random directions and at a distance $r$ in m, estimated by a Gaussian and negative exponential (fat-tailed) dispersal function (Eq. 5, Kruse et al., 2016):

$$r = \sqrt{2E_0^2\left(-log(rand)\right) + \frac{1}{2}distanceratio \cdot rand^{-1.5}} \tag{4}$$

where $E_0$, originally named *width*, is the Gaussian distribution's standard deviation in m, $rand$ stands for a random number $\in [0,1]$ and $distanceratio$ is a weighing factor for the fat tail in m$^2$. Parameter estimates were based on a sensitivity analysis in Kruse et al. (2016) and numerical experiments.

The wind-dependent distance estimation was implemented as a ballistic flight following the assumptions of Matlack (1987). Accordingly, seed dispersal distances depend on the height of the releasing tree top $H_t$ in m, currently estimated as 75% of $H_t$ (factor $f_{Ht} = 0.75$), and are modified by wind speed $V_W$ in m s$^{-1}$ and a species-specific fall speed of propagules (seed plus wing) $V_d = 0.86$ m s$^{-1}$ for *L. gmelinii*:

$$E_0 = f_{Ht} \cdot H_t \cdot \frac{V_W}{V_d} \tag{5}$$

Finally, the direction for the seed dispersal is determined by wind direction, which was randomly selected from a set of observations (see Section 2.2.5 for details).

### 2.2.3. Parameterisation to fit field data

The model's parameters had to be revised after implementing the model extensions to achieve simulated tree densities comparable to field data. Forest inventory data were recorded for each larch individual with explicit positions on plots of a minimum area of 20 x 20 m for several locations along a density gradient from single-tree stands in the north to dense forest tundra stands in the south visited on summer expeditions in the years 2011 and 2013 in north-central Siberia, Russia (Wieczorek et al., 2017). We conducted simulations on 100 x 100 m areas with closed boundaries initialised by introducing 1,000 seeds in the first 100 years of a stabilisation period of 1,000 years, with forcing climate data randomly sampled from the available data. For the final 80 years of each simulation we used the climate series from the corresponding field site (TY04, see 2.2.4 for details). We visually compared the number of trees at year 2011 from the central 20 x 20 m area to the field survey data, which was the first year of fieldwork. The parameters were manually tuned and we iteratively performed simulation runs to improve the simulation results until finally achieving similar stand densities (numbers of trees) as observed (data not shown; parameter values in Table 1).

### 2.2.4. Temperature and precipitation

Simulations are forced with monthly mean temperature and precipitation sum series from the CRU TS 3.22 database (Harris et al. 2014). These are used to estimate long-term responses and derive the auxiliary climate variables active air temperature (sum of temperatures above 10 °C, AAT10) and vegetation length (number of days exceeding the freezing point, net degree days, NDD0) to calculate tree growth, estimate individual tree mortality and establishment from seeds (details in Kruse et al., 2016). We selected a grid box intersecting a location with a known northern taiga tree stand (CH06 at 70.66° N; 97.71° E, site CF in Wieczorek et al., 2017) and a northern forest tundra stand (TY04, 72.41 °N; 105.45 °E, site FTe in Wieczorek et al., 2017). From the available data we excluded years before 1934, because of missing climate station data and hence unreliable extrapolations in the data set (Mitchell et al., 2004). Furthermore, the final year was set to 2013, which is the latest year of fieldwork. The climate at these sites either allows strong tree growth with mean July temperatures of 13.50 °C, coldest temperatures during January of -33.24 °C and a precipitation sum of ~328 mm per year or only sparse stands to emerge with temperatures of 13.11 and -36.07 °C in July and January, respectively, and ~247 mm annual precipitation (cf. Kruse et al., 2016).

### 2.2.5. Wind speed and direction

The model is driven with pairs of wind speed in $m\ s^{-1}$ and wind direction in degrees [°]. The winds at 10 m above the surface for the years 1979–2012 at 6 hourly resolution were extracted from the ERA-Interim reanalysis data set (Fig. 3; Balsamo et al., 2015). Because of the coarse spatial resolution (80 x 80 km), we considered only the grid box over the climate station Khatanga, which is situated roughly in the centre of the treeline ecotone on the southern Taymyr Peninsula (71.9° N; 102.5° E; Wieczorek et al., 2017). During simulation runs, values are randomly drawn from the year's vegetation period (May to

August; Abaimov, 2010) for each seed dispersal event and for the determination of pollination. For simulated years in which climate data are available but no corresponding wind data, a year is randomly selected.

## 2.3. Sensitivity analyses for dispersal processes

To test the influence of the paramterisation of the variables from the newly introduced functions on the model's results, we ran local sensitivity analyses (Grimm & Railsback, 2005, Cariboni et al., 2007). For each simulation repeat the input parameters (Table 2) were changed by 5 and 50% and a sensitivity value calculated by comparing the results with the reference run:

$$S_{+/-} = \frac{\frac{V_{+/-} - V_{REF}}{V_{REF}}}{\left|\frac{P_{+/-} - P_{REF}}{P_{REF}}\right|} \tag{6}$$

where $V$ is the variable of interest derived from each simulation run and $P$ is the parameter of interest, both plus (+) and minus (-) 5% of the estimated parameter, or with the reference value (Kruse et al., 2016).

The simulations were carried out on hypothetical north-south transects with a width of 100 m and length of 1,000 m using the new model version and allowing seeds to be dispersed along the meridional borders. Populations were initiated on empty areas only in the lowermost 100 m wide and 100 m long area by randomly distributing 1,000 seeds during the first 10 years of a 1,000 year long stabilisation period. During this phase, seeds exceeding the lowermost 100 x 100 m area were removed from the simulation. In the following simulation period seeds could enter the area above 100 m and colonise this empty area. The simulation model randomly drew weather conditions for each year from the complete available period 1934–2013 during the stabilisation and simulation period. These simulations were repeated 30 times and the positions of each individual tree were recorded at the end of the simulation (500 years). To directly compare results from simulations with changed parameters to reference runs the simulation period was repeated for each parameter variation starting with an identical state of the simulation at the end of the stabilisation period and using the same climate series.

For the evaluation of migration rates we selected three target output variables for the area ahead of the 100 m initialisation area: (1) *stemcount* is the total number of stems (trees with a height above 130 cm), (2) *forested area* is the area covered with >100 stems ha[-1], and (3) *peak recruit position* is the position of the maximum number of stems on the basis of a running mean with a 50 m window. Additionally, the variable *stand density*, which is the number of stems in the 20 x 20 m plot in the centre of the lowermost area, was selected to assess impacts on plot level. Furthermore, the *pollination distance* expressed as the mean distance between the pollen-donating and seed-producing trees was calculated for the evaluation of the pollination function. The resulting sensitivity values were tested for significant changes from the reference results (mean of 0) with a t-test with a confidence level of 95%.

### 2.4. Model-performance experiments

The memory load was estimated by adding up the size of all data types within each handled structure simulating a plot of one hectare (Table S1). These were multiplied by the actual number of elements in each of the structures. We calculated mean values of the number of handled items of the final 80 years of the simulations for the evaluation of dispersal processes to estimate the total memory needed for the arrays of trees and seeds and the grid representing the environment (Kruse et al., 2016).

To reduce the computation time, we parallelised the code for estimating pollination probabilities, seed dispersal, and tree density computation of the model using the OMP-library and conducted simulations using 1, 4, 8, and 16 CPUs. The performance of the model was evaluated by recording the computation time of each single simulation year for complete simulation runs (1,080 years). We conducted four different runs, one with only wind dispersal of seeds (SEED), one with seed and pollination (+POLL), and two different parallelised pollination computations. First, we tried to simply compute equally sized parts of the complete list of tree individuals including trees that have not produced seeds on the selected number of CPUs (+POLL_PAR-A). In a second variant (+POLL_PAR-B), we attempted to decrease the potential computational overheads of idle CPUs that had finished their job faster because of fewer individuals that needed to estimate pollination for produced seed's ovules, by cutting the list to only trees that produce seeds. The computation time increases with the actual number of trees and seeds present in simulations. In consequence, we analysed the dependency between the time needed for each simulated year and the number of trees and, additionally, the number of produced seeds by generalised nonparametric regression (using the "gam"-function in R-package "gam"; Hastie, 2017). The dependent variable time $t$ was log-transformed prior to analysis. The explanatory variables – number of trees $Nt$ and seeds $Ns$ – were non-parametrically fitted and tested for non-linearity by comparing the deviance of a model that fits the terms linearly with a chi-squared test. In the initial model formula, we also included the interaction between the explanatory variables and excluded non-significant terms from the linear model (p>0.05) until yielding the final best model.

## 3. Results

### 3.1. Verification of wind-dependency

The simulated seeds were solely dispersed in a north or south direction in coherence to the forcing winds (Fig. 2, Table S2 and S3). The median seed dispersal distances were ~12.2 m with a north wind and ~12.0 m with a south wind with a majority of 95% falling within ~43 m of the seed tree, but with rare (~0.1%) dispersal events >1,000 m (Fig. 2). The distance is equally highly correlated with the release height for both wind directions (*rho*=0.63, p<0.0001; Fig. 2).

The pollination events were mainly coming from the direction of the forcing winds: however, ~18% deviated from the forcing wind direction (Table S4). This variance is introduced by the formulae used for calculating the pollination probability for each seed's ovules on a tree and is further increased by the random selection of a father out of a subset of all possible mature trees

based on the probability density function. The median distance along forcing winds of ~38 m is, in general, shorter by ~3-5 m than in other directions (Table S4).

In north-central Siberia, the main wind directions observed during the vegetation period are a combination of both west and east (Fig. 3, upper row). In some years, one of these directions predominates, and is also characterised by stronger wind events. Accordingly, simulated seeds are dispersed into the general direction of the forcing wind data (Fig. 3, middle row). Dispersal distances can reach up to a maximum of several thousand kilometres, yet the majority of seeds fall within a few hundreds of metres, and these are dispersed over distances depicting the wind speeds as well.

The median pollen flight distances are generally larger than the seed's, with a technically fixed maximum of about the distance from the central plot to the borders (Fig. 3, lower row). Similar to seed dispersal, pollination follows the wind directions and fathers are positioned in the upwind direction of the main occurring winds.

## 3.2. Sensitivity analyses for implemented dispersal processes

The sensitivity analyses for the implemented seed dispersal function was extended for further model parameters that have an influence on the migration rate. In general, the four target variables have the same response direction towards changes in the parameters (Table 3). The stronger the changes, the more apparent becomes the change in the result so that the significance increases strongly from only 25% to 79%. The sensitivity values were of the same order of magnitude with the extreme values of -1.89 and 3.26 for each percent change in the input parameter. Most sensitive is the position of the peak recruitment for the observed migration rate (mean absolute sensitivity of 1.09 and 0.92 for 5 and 50%), whereas the impact on the stand level is of minor importance (with sensitivities of only 0.28 and 0.19). The factor of seed productivity $f_S$ and the influence factor of weather on germination rate $f_{Weather\ Germination}$ led to strongest advances of the *peak recruit position* if increased, which the seed mortality rate on trees $P_{Seed\ Mortality,\ Cones}$ caused when lowered.

The sensitivity values for resulting pollination distances for varied parameters were an absolute mean of change of 0.11 for 5% and 0.02 for 50% with extremes of -0.08 and 0.30 (Table 4). The stronger the change, the more apparent is a change in the results (40 to 70% significant values), although the direction of the changes was similar. However, the change is a magnitude smaller when changed by 50% but the directions were consistent with those expected, and increasing Gregory's $m$ led to farther pollination distances and vice versa for pollen descending velocity $V_{d,\ Pollen}$. The maximum sensitivities increased from -0.07 to 0.09 on the southernmost plot to about -0.11 to 0.14 for the northernmost plot where a higher proportion of significant values could also be observed.

### 3.3. Model performance

#### 3.3.1. Memory consumption

The dynamic arrays need 120 bytes for each tree and 98 bytes for each seed. A further 54 bytes are needed for each of the environmental map tiles and another 117 bytes for the storage of output variables for each simulated year (Table S1). The constant containers use 390 bytes for the weather list and the parameter structures contain 642 bytes. On the basis of a simulated typical dense forest with ~92,000 seeds and ~25,000 tree individuals stored in the structures for each hectare, a simulation will need roughly ~15 MB of RAM in a setup of a 1,000 year initialising phase and a subsequent 80 year simulation phase.

#### 3.3.2. Computation time

The simulation time increased with the number of trees in the simulation and for the contrasting simulation setups – either only wind-dependent seed dispersal SEED or also with the calculation of pollination +POLL (Fig. 4). The generalised additive model, including the number of seeds, explained the increase in computation time best and had the lowest AIC value among all simulation types (Table S6). All incorporated variables, namely number of trees and number of seeds, significantly explained the computation time. The number of trees is the most important explanatory variable at ~79.0%, followed by an interaction term of the number of trees and seeds at ~14.6%, and number of seeds at ~4.4% and a residual of ~2.0% unexplained variation.

Without including the pollination events, the computation takes ~0.6 s to calculate a year of a simulated plot on which 30,000 to 40,000 tree individuals are present (Fig. 4). In contrast, this increases to ~120 s $yr^{-1}$ for a similar stand when calculating the pollen donor for each produced seed (+POLL). The first implemented parallelisation of the pollination process (+POLL_PAR A) shortened the computation time by roughly half to ~65 s $yr^{-1}$ when using eight cores. The second variant (+POLL_PAR B) outruns the first when using one to four cores by a factor of ~4, but did not decrease the computation time significantly when using more cores than four. The increase to 16 CPUs led to a further decrease of computation time only for the first variant.

### 4. Discussion

The assumption of unlimited seedbeds – allowing species in models to grow as soon as climate space allows them – causes high uncertainty in future predictions with dynamic global vegetation models (e.g. Midgley et al., 2007; Neilson et al., 2005; Sato and Ise, 2012). Implementing time-lagged responses in such models highlighted the need for a proper understanding and implementation of processes that limit species' migrations (Snell, 2014; Snell and Cowling, 2015). To reveal and understand the underlying processes that cause time lags, we designed the model LAVESI that represents all life-cycle stages of larches in high detail from seeds to mature trees, producing seeds themselves, which are then distributed in the environment (Kruse et al., 2016). We built this model to simulate responses of the Siberian treeline ecotone, which is solely covered over vast areas by a single tree species of the genus *Larix*. Here we describe the model enhancements to achieve, for the first time, a coupled implementation of wind-driven seed dispersal and pollination in the larch forest simulator LAVESI.

## 4.1. Wind-dependent seed dispersal

The simulated seed dispersal strictly followed the wind forcing and seeds settled in a downwind direction as expected, and not, as in the original model, in a purely ballistic manner (Kruse et al., 2016). We tested in a local sensitivity analysis the influence of different parameters on the stand level and the migration process. Sensitivity values were generally low with mean values between ~0.2 to 1.1, respectively for the stand level and for the migration rate. They are smaller compared to other parameters found in the sensitivity analysis of the first version of LAVESI by Kruse et al. (2016). In accordance with these findings, the new model is more sensitive to changes in parameters at transient stages such that higher values are found for the peak recruit position. Furthermore, only strong changes by 50% led in many case to significant changes in the results, strengthening the robustness of the model to the parameterisation. Those parameters leading to more available seeds and higher proportions of recruits (seed production rate, germination) had the highest sensitivity values and if increased they led to a faster migration and stand infilling. As expected, the parameters seed release height, wind speed, distance ratio, and fall speed of propagules became significant for the migration rate but not for the local stand development. Seed dispersal is dependent on the release height of the seed (Matlack, 1987), which is low in the focus region (Wieczorek et al., 2017) and thus leads to low dispersal distances, compared to other taxa (González-Martínez et al., 2002, *Pinus*, 2006; *Picea*, Piotti et al., 2009). When winds are constantly blowing at 2.78 m s$^{-1}$ (10 km h$^{-1}$), simulated distances seldom reach more than ~43 m and only on very rare occasions are they observed with distances exceeding 1,000 m. The dispersal kernel can thus be described as a combination of a Gaussian distribution, with its maximum fraction reaching ~12 m metres from the releasing tree, and a long tail, best described by an exponential function. This aligns well with the implemented function in the model LAVESI (see details in Kruse et al., 2016). The model results are similar when driven with quasi-real wind data from the reanalysis data set ERA-Interim. The short seed dispersal distances depict well the generally observed values of other larch species. For example, Duncan (1954) found for *Larix laricina* in the northern USA that 94% of seeds fell within 18 m of the releasing trees. Furthermore, Pluess (2011) found in dense forests of *Larix decidua* in the Swiss Alps an effective seed dispersal distance of 2–48 m. Moreover, the directions are now more realistically represented and follow the predominant west and east winds as expected (Fig. 3).

The use of winds from only the vegetation period might have introduced a bias, but it is based on the observation that this is the time when seeds are primarily dispersed (Abaimov, 2010). However, secondary dispersal by winds, due to uplift in strong winds, or travel in winter on frozen surfaces over long distances (Nathan et al., 2011a; cf. Pluess, 2011), or due to wind-independent animal-mediated zoochory (Evstigneev et al., 2017), is currently not represented but could facilitate the migration into tundra further. When applying this model over historical periods, which are not covered by observations, one must be careful as the wind regimes could have shifted their main wind direction from the past to the current setting and might even change in the future (Lisitzin, 2012; Trenberth, 1990). A change, for example, from north–south to the current east–west wind directions could have limited the recent potential migration rate. This could explain the slow response of the treeline in northern Siberia to global warming, in addition to the long life-cycle of larches, as well as prevailing seed limitation in the north (Kruse et al., 2016; Wieczorek et al., 2017).

## 4.2. Pollination coupled to prevailing wind conditions

Pollen dispersal functions are frequently used to reconstruct vegetation composition from palaeo archives, for example in the Landscape Reconstruction Algorithm by Sugita et al. (2010), whereas other models have been used to track pollen clouds in tree stands (review in Jackson and Lyford, 1999; Prentice, 1985). Calculating every pollen dispersal event for each tree and seed is computationally challenging, but it can be simplified following the assumptions of Kuparinen et al. (2007). Hence, we implemented a density-dependent probability function and found in the sensitivity analysis that the pollination process was less affected by changing the input parameters than by the seed dispersal process. Values reached only a mean of ~0.02 when changed by 50% and increased from south to north, which covers a density gradient. Pollen influx from farther distances is more apparent in the more open stands, which furthermore is supported by the findings in the sensitivity analysis of the original model (Kruse et al., 2016). The pollination distance increases linearly with Gregory's $m$, which increases the probability for farther standing trees, and decreases as expected for higher pollen descending velocities $V_{d,\ Pollen}$. The density-dependent probability function assigns pollen donors mostly in an upwind direction, but also has a small angular scattering, which was introduced by the use of the von Mises distribution to capture the stochasticity of this process (Gregory, 1961; Kuparinen et al., 2007). This uncertainty could lead to an overestimation of pollination distances, but this seems unlikely because simulated pollen are travelling distances of ~38 m, which is longer compared to seeds and which is in concordance with observations (e.g. Pluess, 2011). However, the pollen amount and thus the probability of a distant tree to reach a seed-producing tree is dependent on the available resources and could further influence the resulting pollination distances. This relationship was not explicitly included but is partly covered by the use of the tree top height and from this, better performing trees have a higher pollination probability. The pollination distance often reaches the maximal possible distance between two trees in our simulation setup, which was the diagonal of the simulated area. These hypothetical pollen grains could probably reach distances of several hundred metres to kilometres, which would be in the range of the general dispersal distance observed in larches (Dow and Ashley, 1996; Hall, 1986).

## 4.3. Model performance

The individual-based approach of the model LAVESI-WIND, with the extension of wind-dependent seed dispersal and pollination, bears a high potential of knowledge gain, but this comes with some challenges: (1) repeated calculations for millions of individuals (seeds and trees) are computationally intense (e.g. Snell et al. 2014, Svenning et al. 2014, Nabel 2015), and (2) they require a certain amount of memory during the simulation runs. Whereas the memory during each simulation run could be minimised to the needs of the simulation setup, the computational power was historically the limiting factor (Grimm and Railsback, 2005). But with the development of recent computer clusters with hundreds of CPUs, it seems very likely that one can overcome this, allowing us to use detailed and spatially explicit models at a regional scale (e.g. Paik et al., 2006; Zhao et al., 2013).

### 4.3.1. Memory consumption

We estimated the requirements for a hectare of a dense simulated forest as 15 MB of RAM. This means, on typical computer servers, even broad-scale simulation runs are easily feasible for 5,000 x 5,000 m, which would need ~38 GB RAM and take approximately 40 hours. The current LAVESI-WIND version was not fully optimised to lower the needs of memory and many variables that might not be needed for a specialised simulation experiment could be excluded. Although, the original simulation program was not intended to be run over continuous square kilometres of forests (Kruse et al., 2016), this is already possible with the current version. The programming language C++ and the process-based structure of the code support an easy and fast forward development of this model.

### 4.3.2. Computation time needed for millions of trees, seeds and pollination

The computational effort of pollination for each seed's ovule increases with the number of mature trees present on a simulated plot. Therefore, to allow simulations to be run on standard computers in manageable time, it was a major goal to minimise the time needed for each simulated year. To meet this requirement, we parallelised parts of the program code that are computationally intensive, namely the processes of pollination and seed dispersal. With our approach, we have been able to decrease the time so far by a factor of two when using 8 CPUs, in comparison to using only one. Still, overheads from using a standard template library (STL)-list container lead to a negative exponential progression of the computation time needed per year rather than linear improvements (Fig. 4). Additional gains for other not yet parallelised processes are much smaller than these, but there is further potential to reduce the computation time by using different implementations of the parallelisation.

### 4.4. Potential model applications

The new model version LAVESI-WIND allows for the evaluation of the importance of driving processes, which determine the response speed of tree stands growing at the treeline in Siberia. It can therefore be used for a very detailed evaluation of intra-stand processes determining migration speeds and help to improve abstract dynamic global vegetation models (e.g. Sato et al., 2007; Sitch et al., 2003), forest landscape models (e.g. Seidl et al, 2012), or regional forest gap models (e.g. Brazhnik and Shugart, 2015, 2016). Such a detailed representation of forest stands, as in the model presented here, is unlikely to be able to simulate forest dynamics on a continental to global scale (cf. Neilson et al., 2005). Nonetheless, the model can be used to parameterise dispersal kernels constraining inter-grid cell migration in DGVMs (Snell, 2014; Snell and Cowling, 2015). This could be achieved by comparing the migration rate in a continuous landscape in LAVESI-WIND, which covers grid cells of the DGVM to achieve a better representation of processes constraining or enhancing the spread of a plant species (cf. Lehsten et al., 2018). With this new model version, we can approach novel research questions, such as "Do wind regime shifts explain faster or slower migration rates in past climate changes?" Furthermore, one could test how different treeline types determine the migration behaviour in changing environments. These can vary widely, based on the treeline type, being abrupt or with stand densities decreasing with the abiotic gradient and might further be influenced by shrubs that respond faster to current climate warming (e.g. Frost and Epstein, 2014), but which are not represented in the model yet. In addition, this may be influenced by single-tree stands growing ahead of the migration front (Holtmeier and Broll, 2005). Further interesting

questions could be addressed, such as the role of refugia during past glacial periods and their influence on present-day tundra colonisation by trees (Wagner et al., 2015), with a simplified and thus computational effective approach. This is a necessary step because the current model version is computationally to demanding to track the full genealogy over simulated areas and time periods. Upscaling approaches could decrease generally the computation time and allow to expand the simulation over larger areas (e.g. Nabel, 2015, Epstein et al., 2007), however, the individual genetic information that passes thorough the landscape would be lost, which might be of interest. By connecting the borders of a simulation plot along the meridional borders we already implemented boundary conditions that allow the simulation of south-to-north transects, which are representative of the treeline area where highest tree densities occur in the south and treeless areas in the north. Thus, with this model, past migration corridors and timings can be revealed by a landscape-scale simulation, potentially answering important questions of the past biogeography of larch species in Siberia.

Before applying this new model version, however, a proper parameterisation is necessary. Because pollen productivity and pollination distances as well as seed dispersal distances are not yet available for forests of the northernmost treeline area, the next important step would be to evaluate the modelled seed dispersal and pollination processes with field-based data, and finally, to apply this model to achieve realistic predictions of a future treeline. Molecular methods can help to improve the seed dispersal function, especially microsatellite markers, which can uncover connections among subpopulations and even kinships by parentage analyses at the stand level, which would make the effective seed dispersal distances directly inferable (Ashley, 2010; Dow and Ashley, 1996; Piotti et al., 2009; Pluess, 2011). Additionally, these methods can be used to estimate the fat tail of the dispersal function indirectly (Piotti et al., 2009).

Another interesting application would be to use this model to estimate the pollen influx in lakes (cf. Sugita, 2007). Pollen influx rates are widely used for vegetation reconstructions at the tundra-taiga transition zone (e.g. Klemm et al., 2016) and could now be used either to tune the dispersal parameters for a more precise population dynamics prediction, or inversely, to reconstruct ancient tree stands by simulations. Before the genetics or the influx rates are included in the model, however, a revision of boundary conditions for pollen in the model is necessary. This must include a relevant source area for the pollen (cf. Sugita, 2007) to determine to what extent genetic traits are delivered by pollen from beyond the borders of the simulated area. If this can be efficiently parameterised, the model could further be used to track genetic lineages in time.

## 5. Conclusions

We conclude that it is feasible to implement wind-driven seed dispersal and pollination in an individual-based model, which is then able to run across broader areas. However, the simulated area and duration of the simulation are constrained by available computer power and memory, and thus further effort is needed to minimise the computational load of this model in order to allow landscape-scale simulations on a standard computer. With the new model setup, further applications in combination with the genetics of the represented species are now feasible and can bring us detailed knowledge about the behaviour of the treeline and the biogeography of larch species through time.

## 6. Code availability

The source code of the host model is available at GitHub https://github.com/StefanKruse/LAVESI/releases/tag/v1.01, and stored in the zenodo database http://doi.org/10.5281/zenodo.1155486. The updated version presented here is named LAVESI-WIND and the first version 1.0 is accessible at GitHub at https://github.com/StefanKruse/LAVESI/tree/v1.0 and stored at http://doi.org/10.5281/zenodo.1165383.

## 7. Author contributions

S.K. and A.G. planned the study, N.K., A.G. and S.K. updated the model and implemented new functions. A.G. and S.K. performed the simulations and the statistical analysis. S.K. and A.G. wrote the manuscript. U.H. provided substantial advice in the process of data analysis and paper writing.

## 8. Competing interests

The authors declare no competing interests.

## 9. Acknowledgements

We acknowledge Sven Willner for valuable advice in the process of parallelising the program code and Cathy Jenks for proofreading and improving the manuscript. Furthermore, we particularly thank the handling topic editor Hisashi Sato as well as Julia Nabel and one anonymous reviewer for their valuable comments on the previous version of the manuscript. The position of Stefan Kruse is funded by the Helmholtz Initiative Fund.

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

## 11. Figures

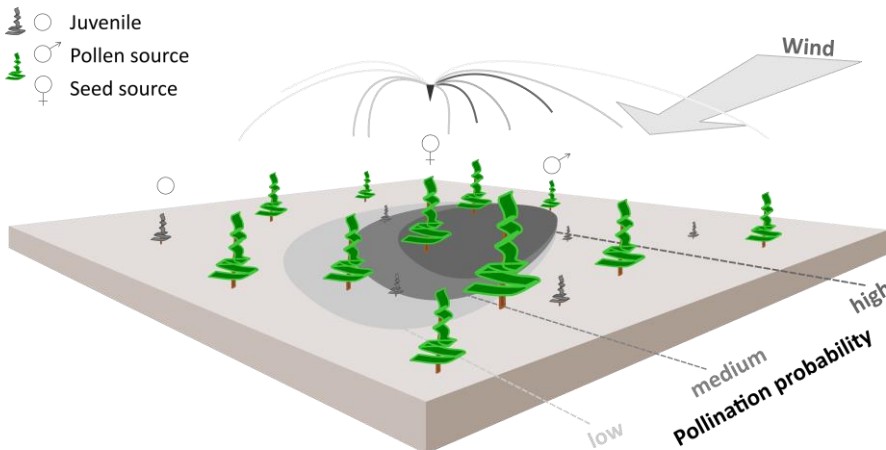

Figure 1. Schematic representation of wind pollination as newly implemented in the LAVESI-WIND model. Based on actual winds, a distance-dependent pollination probability of ovules is estimated for each adult tree (potential pollen source) and for each seed source in the simulated area. The shaded areas on the ground represent the pollination probability for the labelled seed source for winds from the upper-right corner. These are generally higher for adult trees in upwind direction of the central seed source.

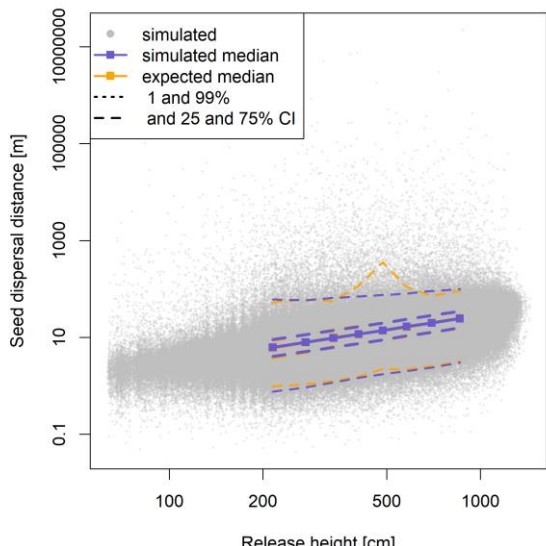

Figure 2. Dispersal distances of seeds are wind dependent and positively correlated with the height of the releasing tree. The simulated and hypothetically calculated dispersals were compared across evenly distributed height classes; the results are similar for north and south winds, and here the results with north winds are presented.

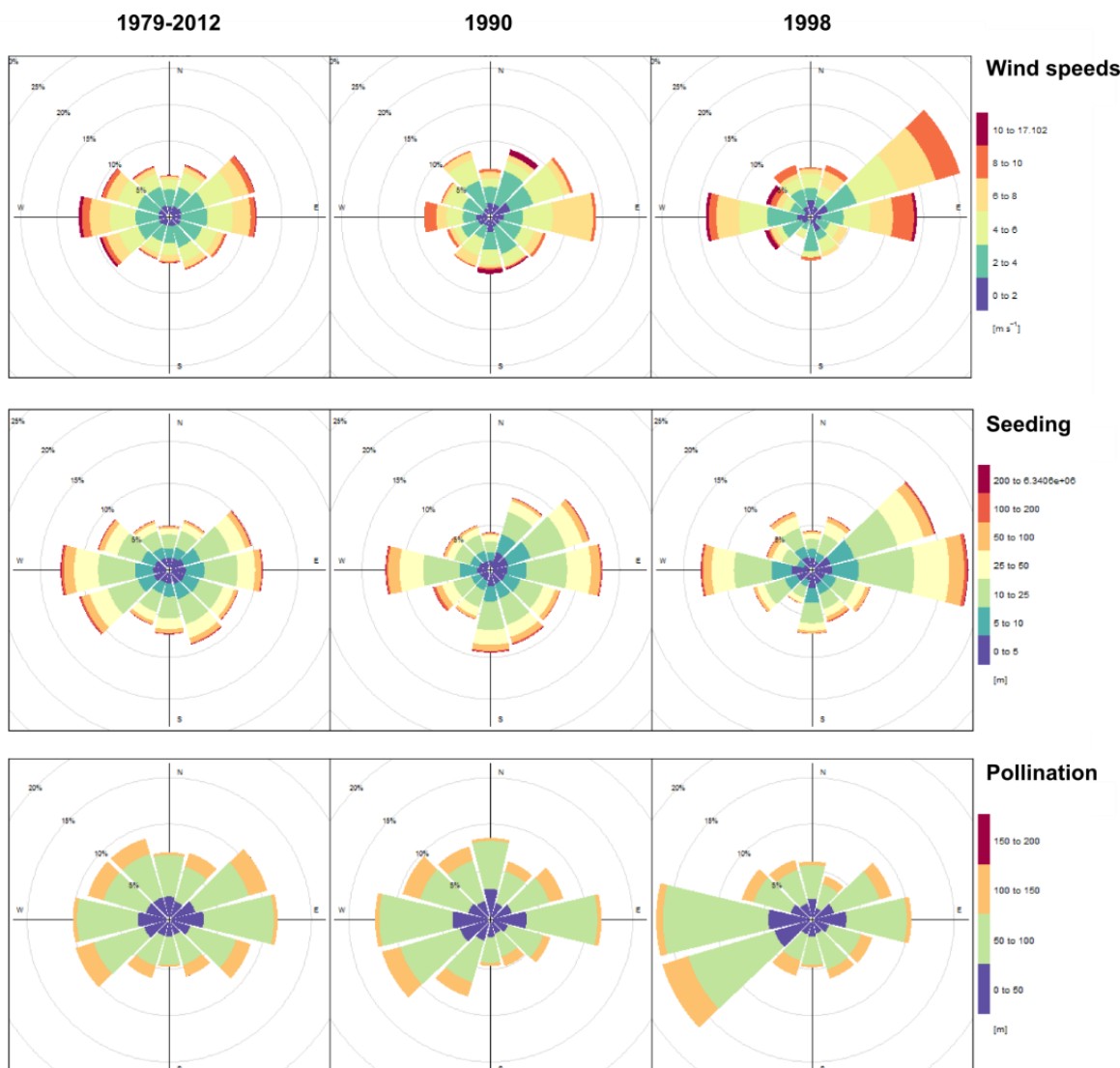

**Figure 3. Wind forcing (upper row), simulated seed dispersal (middle row), and pollination distances (lowest row) by distance and cardinal direction. Simulations were performed on 200 x 200 m plots and seed dispersal events tracked away from source trees: pollination events were recorded from pollen donor trees standing in the plot area into the central 20 x 20 m plot.**

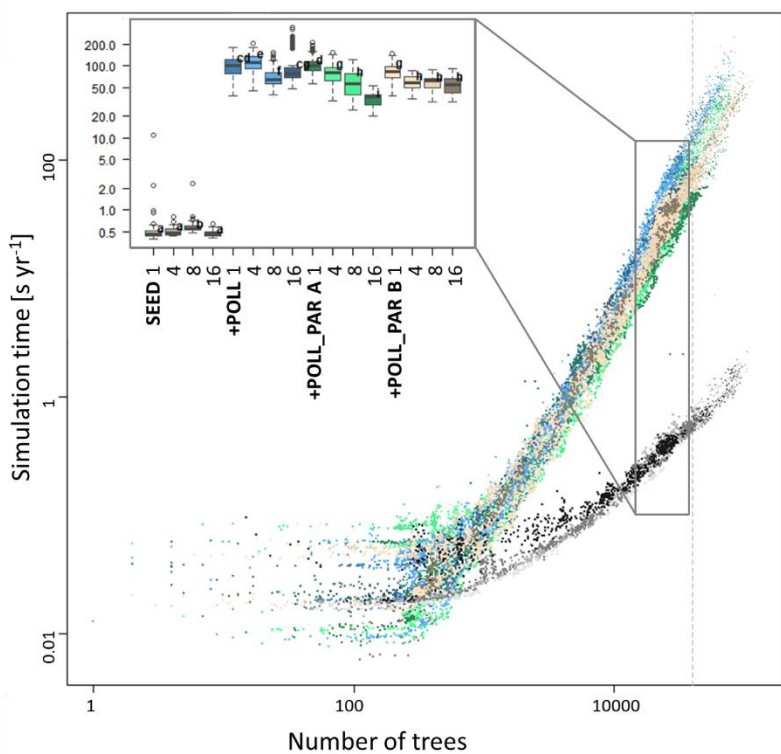

**Figure 4. Simulation consumption time in relation to the number of trees present, the number of CPUs used, and for different types of parallelisation of the code. The time increases exponentially with the number of trees and more quickly when simulating the additional pollination (+POLL) compared with just the explicit seed dispersal (SEED). The inset summarises the simulation time for simulated typical northern taiga stands, ranging between 30,000 and 40,000 trees. The letters next to the boxes indicate similar groups inferred with a Wilcoxon-test and Holm correction for multiple testing.**

## 12. Tables

**Table 1. Overview of model parameters and processes for *L. gmelinii* individuals that are different from the original version (Kruse et al., 2016).**

| Parameter | Value and dimension | References |
|---|---|---|
| **Growth** | | |
| Quadratic term of the equation for diameter growth rate | -0.003 ln(cm)/cm² | data-based estimate similar to Fyllas *et al.* (2010) |
| Linear term of the growth function | 0.030 ln(cm)/cm | |
| Constant term of the growth function | -1.98 ln(cm) | |
| **Seed production, dispersal and establishment** | | |
| Factor of seed productivity | 8 | literature-based estimate (Kruklis & Milyutin, 1977, cited in Abaimov, 2010) |
| Background germination rate | 0.01 | tuned |
| Horizontal seed dispersal distance at wind speed of 10 km/h | 60.1 m | estimated after Matlack (1987) |
| Seed descent rate | 0.86 m/s | estimated descent rate based on Matlack (1987) |
| **Mortality** | | |
| Background mortality rate | 0.0001 $yr^{-1}$ | data-based estimate |
| Current tree growth influence factor on tree mortality | 0.0 | tuned |
| Weather influence factor on tree mortality | 0.1 | tuned |
| Density influence factor on tree mortality | 2.0 | tuned |
| Seed fertility | 2 years | Ban *et al.* (1998) |
| Mean temperature of the coldest month (January) at the border of the species' geographical range | -45 °C | Shugart *et al.* (1992) |
| Exponent scaling the height influence | 0.2 | tuned |
| **Weather processing** | | |
| Exponent scaling the influence of surrounding density for a tree | 0.1 | tuned |
| Exponent scaling the density value | 0.5 | tuned |

**Table 2. Parameter values evaluated in the sensitivity analysis for seed dispersal, migration patterns, and pollination.**

| Parameter | Reference value and dimension |
|---|---|
| **Seed dispersal function** | |
| Maximal flight distance for *L. gmelinii* seeds at 10 km h$^{-1}$ ($r_{Maximum\ Seeds}$, Matlack, 1987) | 60.1 m |
| Species-specific fall speed of propagules ($V_d$) | 0.86 m s$^{-1}$ |
| Distance ratio weighing factor ($sdist$) | 0.16 |
| Factor of seed productivity ($f_S$) | 8 |
| Background germination rate ($f_{Background\ Germination}$) | 0.01 |
| Influence factor of weather on germination rate ($f_{Weather\ Germination}$) | 0.447975 |
| Maximum age of seeds ($age_{Maximum\ Seeds}$) | 2 yrs |
| Seed mortality rate on trees (in cones, $P_{Seed\ Mortality,\ Cones}$) and | 0.44724 |
| at the ground ($P_{Seed\ Mortality,\ Ground}$) | 0.55803 |
| Factor for release height estimation $H_t$ ($f_{Ht}$) | 0.75 |
| Factor for the actual wind direction $\overline{\theta}$ ($f_{\overline{\theta}}$) | 1 |
| Factor for wind speeds $V_w$ ($f_{Vw}$) | 1 |
| Probability of seed release from cones ($P_{Seed\ Release}$) | 0.63931 |
| **Pollination** | |
| Inverse of the von Mises distribution's variance ($\kappa$) | 10 |
| Gregory's parameter $C$ | 0.6 cm$^{-(1-0.5m)}$ |
| Gregory's parameter $m$ | 1.25 |
| Pollen descending velocity ($V_{d,\ Pollen}$) | 0.126 m s$^{-1}$ |
| Factor for the actual wind direction ($\overline{\theta}$) | 1 |

**Table 3. Sensitivity values for varied model parameters influencing the seed dispersal process. Bold values are highly significant with p<0.01, italic with p<0.05, and grey values non-significantly different from the reference run. Values are the mean over 30 simulations.**

| Parameter | Stand density 5% S- | S+ | 50% S- | S+ | Forested area 5% S- | S+ | 50% S- | S+ | Stemcount 5% S- | S+ | 50% S- | S+ | Peak recruit position 5% S- | S+ | 50% S- | S+ |
|---|---|---|---|---|---|---|---|---|---|---|---|---|---|---|---|---|
| $r_{Maximum\ Seeds}$ | **0.6±1.4** | 0.2±1.5 | **0.1±0.1** | 0.0±0.2 | 0.0±3.2 | 0.7±3.1 | 0.0±0.3 | 0.0±0.4 | 0.0±3.0 | 0.3±3.1 | 0.0±0.3 | 0.0±0.3 | 0.9±4.5 | 0.0±4.4 | 0.0±0.4 | 0.0±0.3 |
| $V_d$ | 0.0±1.6 | *0.5±1.7* | 0.0±0.2 | 0.0±0.1 | 1.0±2.7 | 0.0±3.1 | 0.0±0.4 | **0.2±0.4** | 0.3±2.3 | -0.2±2.5 | -0.1±0.3 | **0.3±0.4** | 0.8±3.7 | -0.4±3.8 | **-0.3±0.5** | **1.4±0.5** |
| $sdist$ | 0.0±1.4 | 0.2±1.6 | **0.1±0.2** | **0.0±0.1** | 0.2±2.8 | 0.8±3.4 | **-0.3±0.4** | **0.2±0.4** | 0.0±2.9 | 0.6±3.0 | **-0.3±0.3** | **0.2±0.3** | 0.5±4.3 | 0.9±4.4 | -0.1±0.4 | **0.4±0.5** |
| $f_S$ | -0.2±1.6 | **0.7±1.5** | **-0.6±0.1** | **0.4±0.2** | -0.8±3.3 | **1.9±1.9** | **-1.5±0.1** | **0.4±0.3** | **-1.9±2.7** | **2.9±2.1** | **-1.5±0.1** | **1.7±0.5** | *-1.9±4.5* | **2.9±5.0** | **-1.3±0.2** | **3.0±1.1** |
| $f_{Background\ Germination}$ | 0.2±1.5 | **0.6±1.7** | *0.0±0.1* | **0.1±0.2** | 0.0±3.5 | 0.1±2.7 | 0.0±0.3 | 0.1±0.2 | -0.1±2.8 | 0.2±2.5 | *-0.1±0.3* | 0.1±0.3 | 0.7±4.1 | 0.9±3.7 | 0.0±0.3 | *0.2±0.4* |
| $f_{Weather\ Germination}$ | -0.1±1.6 | **0.6±1.5** | **-0.5±0.1** | **0.3±0.2** | -0.6±3.6 | 0.8±2.8 | **-1.4±0.1** | **0.4±0.4** | -1.5±3.0 | **2.2±2.8** | **-1.5±0.1** | **1.5±0.5** | 0.3±4.3 | *3.3±7.2* | **-1.1±0.2** | *3.0±1.6* |
| $age_{Maximum\ Seeds}$ [1] | | | **-0.4±0.1** | **0.2±0.2** | | | **-1.1±0.2** | **0.2±0.4** | | | **-1.2±0.1** | **0.6±0.4** | | | **-0.9±0.2** | **0.8±0.6** |
| $P_{Seed\ Mortality,\ Cones}$ | **0.6±1.6** | -0.3±1.6 | **0.3±0.2** | **-0.5±0.1** | 0.6±3.0 | -0.4±2.2 | **0.4±0.3** | **-1.3±0.2** | **1.6±2.7** | **-1.5±1.6** | **1.5±0.6** | **-1.4±0.1** | *2.6±5.4* | -0.6±3.9 | **2.9±2.1** | **-1.1±0.2** |
| $P_{Seed\ Mortality,\ Ground}$ | 0.1±1.5 | -0.1±1.3 | **0.2±0.2** | **-0.1±0.1** | 0.1±3.0 | -0.1±3.4 | **0.3±0.4** | **-0.3±0.3** | 0.1±2.7 | -0.5±2.7 | **0.6±0.4** | **-0.5±0.2** | 1.3±4.1 | 0.6±3.5 | **0.6±0.5** | **-0.4±0.2** |
| $f_{Ht}$ | *0.4±1.3* | 0.2±1.7 | 0.0±0.2 | 0.0±0.1 | 0.1±3.1 | -0.1±3.1 | **-0.2±0.3** | 0.1±0.4 | 0.3±2.8 | 0.0±2.8 | **-0.3±0.2** | *0.2±0.4* | 0.0±3.0 | 0.2±3.7 | **-0.5±0.3** | **0.7±0.5** |
| $f_{\overline{\theta}}$ | 0.0±1.4 | 0.2±1.4 | 0.0±0.2 | 0.0±0.2 | -0.4±2.2 | *1.5±3.0* | 0.1±0.3 | **0.3±0.4** | -0.4±2.2 | **1.8±3.1** | *0.1±0.4* | **0.3±0.4** | -0.1±5.7 | **2.9±3.8** | **0.3±0.4** | **0.7±0.5** |
| $f_{Vw}$ | 0.1±1.6 | 0.1±1.7 | **0.1±0.2** | **0.0±0.1** | 0.2±2.9 | 0.2±3.2 | **-0.3±0.3** | *0.2±0.4* | 0.1±2.5 | 0.3±3.8 | **-0.4±0.2** | *0.2±0.4* | 0.3±3.3 | *1.6±3.7* | **-0.6±0.4** | **0.7±0.3** |
| $P_{Seed\ Release}$ | -0.1±1.5 | 0.4±1.7 | **-0.5±0.1** | **0.3±0.2** | -0.1±2.9 | 0.3±3.2 | **-1.3±0.2** | **0.4±0.4** | -0.7±2.5 | *1.4±3.0* | **-1.4±0.1** | **1.2±0.5** | -0.5±3.3 | **2.0±3.9** | **-1.1±0.2** | **1.9±1.4** |
| Mean absolute $S_{-,+}$ | **0.28±0.21** | | **0.19±0.20** | | **0.79±0.82** | | **0.66±0.60** | | **0.46±0.47** | | **0.43±0.46** | | **1.09±0.99** | | **0.92±0.88** | |

[1] The integer variable maximum age of seeds was excluded from the 5% change sensitivity analysis as only 50% changes had valid values.

**Table 4. Sensitivity values of the model's results assessed by mean distance per pollination event into an area of 20 x 20 m in the north, middle, and south of 100-m-wide and 1-km-long transects. Bold values are highly significant with p<0.01, italic with p<0.05, and grey values non-significantly different from the reference run. Values are the mean over 30 simulations.**

| Parameter | Sensitivity | | | |
| --- | --- | --- | --- | --- |
| | 5% | | 50% | |
| | S- | S+ | S- | S+ |
| **North (influx from south)** | | | | |
| $\kappa$ | -0.05±0.33 | *-0.07±0.28* | **0.01±0.03** | *0.01±0.03* |
| $C$ | **0.08±0.31** | -0.02±0.28 | *-0.01±0.03* | 0.00±0.03 |
| $m$ | *0.07±0.28* | **-0.11±0.27** | **-0.03±0.03** | **0.02±0.03** |
| $V_{d,\ Pollen}$ | 0.05±0.31 | **0.14±0.29** | **0.01±0.03** | **-0.01±0.03** |
| $\overline{\theta}$ | 0.05±0.31 | 0.02±0.24 | **-0.01±0.03** | -0.01±0.03 |
| **Middle (influx from all directions)** | | | | |
| $\kappa$ | 0.02±0.54 | **0.17±0.62** | 0.00±0.07 | **0.02±0.06** |
| $C$ | 0.1±0.58 | **0.22±0.6** | -0.02±0.06 | **0.02±0.06** |
| $m$ | -0.04±0.61 | *0.11±0.63* | **-0.03±0.07** | **0.05±0.06** |
| $V_{d,\ Pollen}$ | 0.07±0.69 | -0.08±0.62 | *0.01±0.05* | -0.01±0.06 |
| $\overline{\theta}$ | 0.01±0.66 | **0.3±0.61** | -0.01±0.06 | **0.04±0.06** |
| **South (influx from north)** | | | | |
| $\kappa$ | 0.03±0.43 | -0.04±0.38 | 0.00±0.04 | *0.01±0.04* |
| $C$ | -0.01±0.4 | 0.01±0.37 | 0.00±0.05 | 0.00±0.04 |
| $m$ | -0.07±0.41 | -0.06±0.37 | **-0.01±0.04** | 0.01±0.04 |
| $V_{d,\ Pollen}$ | *0.08±0.39* | *0.09±0.37* | 0.00±0.04 | -0.01±0.04 |
| $\overline{\theta}$ | -0.05±0.36 | 0.06±0.39 | **-0.01±0.04** | 0.00±0.04 |

725