# Peer review of "Implementing spatially explicit wind-driven seed and pollen dispersal in the individual-based larch simulation model: LAVESI-WIND 1.0"

_Geoscientific Model Development, 2018_

## Referee Comment (RC1) · Anonymous Referee #1 · 17 Apr 2018

General comments:
* * *
The authors present approaches how to include wind-dependent seed dispersal and direction-dependent pollination into a mono-species, spatially explicit, small scale forest model. They verify whether the new implementation behaves as expected and evaluate the distance and direction dependent seed dispersal and pollen input probability in comparison to wind direction and speed.

The two new submodels are novel and could also be transferred to other spatially-explicit models. This is the main merit of the study. In particular by the pollination

model the field of forest modelling is substantially advanced.

The manuscript is well structured and in large parts well written. Equations and parameters are given.

Sometimes more explanations would be very helpful, in particular concerning the pollination module.

In the introduction and discussion a lot of weight is put on migration processes and the inability of large-scale models to deal with it. In this context it seems at least a bit strange that the authors restrict their simulation to one plot of at maximum 200m*200m. It would be more convincing to make clear that you started with a detailed small scale model which later could maybe inform larger scale models and in particular can enhance understanding of gene flow in forests.

The question remains: how can this approach be adopted by models running on a larger scale? Can, for example, the detailed simulation of individual seeds and trees be upscaled? Does it make sense to use approaches as discussed in (Snell, 2014, Using dynamic vegetation models) ? Please discuss.

As already pointed out using only a small area of 4 ha for studying migration processes seems weird. Also the circular boundary conditions might confound results of spatial processes because spatial gradients disappear. (Maybe I got that wrong, but then it indicates that a better explanation is needed!)

It also seems to me that ignoring the size or age of father trees, and by that the relative amount they contribute to the pollen cloud, can affect the results.

A lot would be clarified by more detailed captions of the figures.

In general, more clarifications are needed in many parts. All my questions below indicate unclear formulations which should be improved.
* * *
Specific comments
* * *
Throughout the text: avoid the term "pollination of seeds". Biologically the ovules are pollinated, and this yields the seeds. Or at least clarify that you mean the ovules in this context.

L36: There are forest models for all scales, not only for stands. Omit "stand".

L37: Between forest stand models and DGVMs, there are forest landscape models, which incorporate seed (whereas not pollen) dispersal (c.f. , Snell et al, 2014, Using dynamic vegetation models; Shifley et al., 2017) and have been applied also to the arctic (e.g. TreeMig, Epstein et al, 2007) .

L50: what do you mean by "at the molecular level"? The genes? The coded proteins? Or the resulting traits?

L50: Why does high gene flow constrain local adaptation? I would assume that it enhances it. Omit "high", or exchange "constrain" by "influence".

L51: I do not understand how local adaptation prevents negative consequences (which ones?) due to founder effects. Clarify

L63: commonly -> sometimes. E.g. concerning DGVMs, to my knowledge up to now Rebecca Snell's LPJ-GUESS version is the only one with something a bit similar to dispersal. Forest Landscape models of course contain dispersal, but believe it or not, it is very often switched off!

L101: strict = absorbing?

L08: It would biologically be more logical to start with pollination and describe seed dispersal afterwards

L111: the formula is hard to read, one has to guess: (1/2)*distanceratio*(rand)**1.5

???

L116: height of the tree top?

L120: Unclear: Probably you mean: wind speed and direction, which were randomly selected from a set of observations

L121: Start with a general conceptual description of the pollination and refer to fig. 1.

L122: Is the pollination simulated really for each individual ovule? Idea for an upscaling: Would it be possible to do that more in a "share" approach per tree (assuming that all ovules are of the same genotype): Of N seeds of the tree, the share p1 are pollinated with pollen from father 1, p2 from father 2,...

L124: now you confuse me: tree or seed?

L131: make the reference to fig. 1 more prominent, I overread it several times, and found fig. 1 just by chance.

L138: "pollen distribution" , isn't it rather the overall pollination probability of a seed?

L140: Please make explicit if and how the pollen amount comes into play and how it depends on tree size or age or weather. How is a father tree defined by the variables above? How does its distance to the mother tree come into play? I see it in fig. 1, but not in the formulae

L160: runaway line   L187: What is the difference between the two parallelisations A and B?

L189-205: I'm confused. What is the difference between the implemented dispersal function and the full model? Isn't the dispersal function implemented into the full model? So, where can differences come from? Why do you need this verification? Please clarify.

L239: Why do you undergo the effort to calculate the wind-direction-dependent pollen

transport, to then take father trees randomly from any direction???? Why not using the father trees in the different directions? Or did I misunderstand that? Do you really sample the father trees with uniform probability?

L260: Why do you mention the times? Because the RAM adds up over time, or because the numbers of trees and seeds increase over time?

L284: and forest landscape models, in particular individual based ones like Iland (Seidl et al, 2012)

L312: Unclear. What would slow down the recent potential migration rate? And how?

L358: This would require simulations on very large areas or long transects, for which the current approach is (still) too slow. Some simplification/upscaling would be required.

L365-370: This is the real step forward, I would even emphasize it more!

Fig. 3: Please extend the caption, explaining in much more detail what was recorded. In particular the directions need to be explained. I guess, for wind and seeds, the direction means in which direction they go, in contrast for pollination the direction refers to where the father trees are?

#############

Additional literature:

#############

Epstein, H.E., Yu, Q., Kaplan, J.O. & Lischke, H. (2007) Simulating future changes in Arctic and subarctic vegetation. Computing in Science & Engineering, 9, 12-23.) .

Seidl, R., Rammer, W., Scheller, R.M. & Spies, T.A. (2012) An individual-based process model to simulate landscape-scale forest ecosystem dynamics. Ecological Modelling, 231, 87-100.

Snell, R.S., Huth, A., Nabel, J.E.M.S., Bocedi, G., Travis, J.M.J., Gravel, D., Bugmann, H., Gutiérrez, A.G., Hickler, T., Higgins, S.I., Reineking, B., Scherstjanoi, M., Zurbriggen, N. & Lischke, H. ( 2014), Using dynamic vegetation models to simulate plant range shifts. Ecography, 37, 1184-1197)

Shifley, S.R., He, H.S., Lischke, H., Wang, W., Jin, W., Gustafson, E.J., Thompson, J.R., Thompson III, F.R., Dijak, W.D. & Yang, J. (2017) The past and future of modeling forest dynamics: From growth and yield curves to forest landscape models. Landscape Ecology, 32, 1307-1325.)

---

## Referee Comment (RC2) · Anonymous Referee #2 · 19 Apr 2018

The paper presents the wind driven seed and pollen dispersal algorithms newly implemented in the existing model LAVESI which simulates Larix gmelini vegetation dynamics in north-central Siberia. Language wise the paper is very well written. Content-wise, however, I feel that the manuscript is currently a bit thin. I do agree that seed dispersal and particularly pollination are neglected processes worth to be further investigated. New contributions are the insertion of wind speed and distance in the already existing seed dispersal kernel and the new pollination probability function. Beside this, however, it is only shown that the implementation of the functions work as intended and that two unspecified parallelisations lead to a computation time reduction. Else, the authors mainly write what potentially could be done with the model, but unfortunately

miss to give sufficient instructions for how this could be done, nor do they show example simulations for the potential applications.

I do not understand why the authors decided to put so much emphasis on describing the verification with all its sophisticated applied statistics. In my opinion the verification does not contribute to a knowledge gain and could be dealt with a few sentences, referring the main share to the supplementary. However, I do believe that a transparent description of the parameterisation and applied sensitivity analyses could be of large value for the community.

I recommend a thorough revision of the manuscript targeting that in the end only those applications are emphasised in the abstract, introduction and conclusion that are sufficiently prepared in the methods and results and appropriately discussed.

In the following I give examples for recommended revisions in the different sections followed by a list of small comments, questions and remarks.

(1) Abstract: In my opinion the abstract does currently not fit to the content of the paper. Reading it, I developed different expectations not addressed in the manuscript, such as model applications (at least exemplary) dealing with the advocated passing of genetic information and/or diversity and/or identifying important drivers of migration dynamics. Also global vegetation models are mentioned twice prominently in the abstract, e.g. last sentence: '... substantially help in unveiling the important drivers ...', however, unfortunately, I did not get an idea from the manuscript how this could be done and how global vegetation models can be improved. I recommend rewriting the abstract more closely describing the manuscript.

(2) Introduction: The authors refer to DGVMs and list processes not implemented in them. However, they do not mention why these processes are not included – e.g. issues of parameterisation (see e.g. Snell et al. 2014), particularly also due to the high variation in seed dispersal rates (and vectors) within the plant functional types currently used (see e.g. Huntley et al. 2010, Lee 2011, Snell et al. 2014, Svenning et al. 2014), issues of coarse spatial resolution and of computational complexity arising with inter-grid cell communication via seed dispersal or other spatially linked processes (e.g. Nabel 2015) – yet these issues that cause problems for larger scale models seem to even get visible in this manuscript describing a local single species model: when discussing the huge increase in computation cost and that the model is not properly parameterised for the suggested studies (indicated by sentences such as "Before applying this new model version, however, a proper parameterisation is necessary." "If this can be efficiently parameterised, the model could further be used to track genetic lineages in time."). I recommend mentioning these prominent problems of seed dispersal implementation in the introduction and relate to them in the discussion.

(3) Methods: the methods seem largely appropriate. Below I list some specific questions/recommendations:

(a) According to 2.1 the presented simulations assume homogeneous forest stands with periodic boundary conditions. However, applications to study tree line dynamics, or migration in general, or the suggested "impact of migration processes on the genetics", require dispersal among inhomogeneous plots/gridcells and I would appreciate at least a small application showing how the model can be used for such conditions, showing e.g. what migration speed results from the model, which processes have largest influences etc.

(b) "2.2.4 parameterisation to fit field data": even if the authors only visually compared the resulting number of trees, it might be helpful to have a table listing the number of trees for different parameter combinations (at least in the supplementary). What sensitivities were detected, which parameters were particularly important/ influential in tuning the number of trees? The authors state that the model parameters required a revision after implementing the extensions - what can we learn from this? It might be worth to discuss which parameters had to be revised and how, which are sensitive and influential - these things could be really helpful for the community.

(c) Most of the parameters in Table 1 are stated as "estimated" but there at least needs to be a starting point! If the study aims to help parameterising DGVMs it would be helpful to at least know on what the parameters are originally based, even if its from another species etc.

(d) 2.3.2: I find this section difficult to read and understand. If I am not mistaken the dispersal function was once implemented inside and once implemented outside the model (which I think is not explicitly stated somewhere?). I would move most of this and the according results section to the supplementary and only retain that the model was successfully verified.

(e) 2.3.3: The header of this section suggests an evaluation. The text of the section suggests that the simulation results will be quantitatively compared to observed values from literature. Yet I did not find such a comparison in the results section.

(4) Results: thin (see remarks above)

(a) As stated above, I would move most of 3.1 to the supplementary (all comparison to 'direcly' estimated/calculated).

(b) I do not get the "fraction of significant differences" in Table 2

(c) The second parallelisation variant outruns the first when using four cores, however, there seems to be a saturation in that 8 cores are not doing better but even worse than 4, why is this? It would be interesting to see what happens with 16/more cores!

(5) Discussion:

(a) The authors start with stating that unlimited seedbeds cause high uncertainty in DGVM predictions. While I agree that the assumption of unlimited seedbeds contributes to the uncertainty in DGVM studies/projections I think that there are heaps of at least equally important uncertainty sources, like modelling of mortality, competition, canopy structures, etc. Furthermore, I do not see that the manuscript present a remedy for the uncertainty connected to seedbeds. In my opinion a min/max assumption

with no and full dispersal currently gives a better uncertainty frame than difficult to parametrise dispersal kernels. If one of the main messages of the manuscript stays that LAVESI-WIND can be used to enhance DGVMs, e.g. by parameterising dispersal kernels, I strongly recommend to discuss the issue and difficulties of (PFT) parameterisation (see e.g. Huntley et al. 2010, Lee 2011, Snell et al. 2014).

(b) LAVESI is a single species model parameterised for a certain Larix species. I wonder how not represented processes, such as competition with other species/understory, trait variability, or others, could influence migration processes, particularly in the currently tree-less tundra? Is this implicitly covered by the parameterisation? If it's implicitly covered, how might this change with climate change?

(c) Large parts of the model are realised as stochastic processes. How many repetitions are required to get a sound result, e.g. how is a targeted output variable such as migration speed influenced by the stochasticity?

(d) The last sentence of 4.2. stating that the model is not yet parameterisation to field data sounds as if it would only be a small step, but I assume that this is rather difficult and would like to read a bit more about how this can be done and what problems are anticipated.

(e) The increase in computation costs associated with seed dispersal and the like have been mentioned in several places (e.g. Snell et al. 2014, Svenning et al. 2014, Nabel 2015) and are one of the reasons why seed dispersal is usually not accounted for on larger simulation extents. I think it would be worth to mention that this is a common finding when dealing with seed dispersal. Furthermore, the authors mention hundreds of CPUS here but did only conduct simulations with 1,4 and 8 CPUs (and with one of the parallelisation the avg sim time was worse with 8 than with 4 CPUs). In order to better support their statements and in order to assess the merit of the parallelisations another set of simulations with e.g. 16 or 32 CPUs would be helpful.

(f) In my opinion section 4.4 is not well supported by the rest of the manuscript. I for

example miss a link from simulations on the homogeneous 200 x 200 m plot with periodic boundaries to a study investigating migration rates. How are borders of simulation plots connected? How is the long distance dispersal dealt with and pollination which potentially reach over several plots? How does this increase the computational complexity, is such a simulation currently computationally feasible? Especially, when you a certain set of repetitions is required to account for the stochasticity? Furthermore, how can pollen influx be studied when the amount of pollen is not simulated by the model?

=======================================================

small comments, questions and remarks:

l.1 - title: Since LAVESI did already contain spatially explicit seed dispersal, maybe add wind driven "Implementing spatially explicit wind driven seed and pollen dispersal ..."

l.23 - is the selected pollen donor still potential? maybe: "... select the pollen donor."

l.46 - "is mainly caused"

l.47 - "using a time lagged response function"? Snell 2014 uses a dispersal function for dispersal between grid cells and a logistic function to limit within grid cell dispersal"

l.53 - "this" refers to what? local adaptation?

l.59 - "this ensured the most realistic implementation" I would remove this part of the sentence or rephrase the paragraph, because of the previously mentioned dynamics at the treeline. I had to read the sentence several times before I realised that this refers to the homogeneous forest only.

l.63 - "are not coupled" – There is a phd thesis integrating wind dispersal in the DGVM CLM, see https://globalchange.mit.edu/sites/default/files/Lee_PhD_2011.pdf

l.64 - 'Among others, Student's 2Dt...' This sentence seems lost - delete it?

l.69-77: move to methods or discussion

l.84: "Results of the validation" -> verification?

l.83-86: I would delete these sentences, particularly the second part informing that there is a discussion and conclusion section.

l.93: I would use past here (i.e. "which were parameterised"), since you describe the parameterisation it in Kruse et al. (2016) and, if I understood it correctly, re-parametrised it for this study?

l.95: What exactly are "homogeneous forest plots of 100 x 100 m" - 1. does a simulation runs on one plot or several plots? 2. Does homogeneous refer to the environmental conditions? What is the resolution and what is the extent of the simulation runs?

l.95: "seed dispersed beyond the plot borders" - thus out of the 100 x 100m plot?

l.99: "... lag the hypothetical warming ..." – maybe "... densification and northwards migration might lag the applied hypothetical warming ..."

l.111: for better readability maybe insert a space or comma or other separator between distanceratio and rand

l.113: Does this refer to (1) the sensitivity analysis in Kruse et al. 2016 or (2) a new analysis? If (1) maybe point again to the reference, if (2) can you show results of the sensitivity analysis in your supplementary?

l.116+117: do you have references for the 75%, for Vd and lambda? Is there any biological interpretation for lambda?

l.120: a reference to 2.2.5 would be helpful here

l.124: "subsequently" to what?

l.128: could you shortly write what C and m are? Why do you use m=1.25? Gregory (1961) seems to recommend different values for m for different conditions.

l.133: the symbols for the angle between trees and the actual wind direction are very

difficult to visibly distinguish, because the bar is hardly noticeable. Maybe make the bar more visible, lower, or use another symbol for the wind direction

l.143: "monthly climate series" - reading that the model has an annual temporal resolution I had to went back to the LAVESI description. I think it might be helpful to very briefly state which CRU variables are used to derive which annual driving variables

l.147: in this sentence field work is mentioned but it is neither explained what field work this was, nor a reference given. In l.156 I can infer that the reference probably is Wieczorek et al. (2017) and from this and other places I infer that the field work took place from 2011-2013. Maybe it would be easier to swap 2.2.3 and 2.2.4 and shortly properly introduce the field work.

l.149: should probably be "-36°C"

l.153: which field data? (see above). 100 x 100 areas -> is this the one 100x100m plot? How many repetitions did you simulate?

l.155: I do not get which climate you use. Is this the climate described in 2.2.3 or not? If so: do not repeat, if not: why not?

l.156: even if the number pf trees was only visually compared it would be nice to see some combinations of number of trees and parameters in a table

l.158: what is a good fit?

l.160: "Climate forcing data" – leftover?

l.171: do you refer with "plots" to the 50 repetitions of the one simulated plot?

l.182: from all pollen sources or from the selected donors?

l.184: "implemented simulations" -> "conducted simulations"

l.184-: repetition with 225

l.187: How do these two parallelisations differ?

l.203: "north between 345° and 15°" - how does this compare to the directions in Table 2?

l.219: plural vs singular: "computation time increases"

l.220: readability: "produced seeds"

l.225-: repetition with 184 (however without 2 CPUs)

l.239: uniform ? -> 2.2.2: "based on the probability density"

l.242: "the higher values" - which?

l.242-: I do not get: "4-6% of test show significant differences ... no evidence that the differences are significant"?

l.257: Maybe output storage is not in the Table S1?

l.269: maybe replace inferring by calculating/tracking

l.270: maybe replace estimating by calculating

l.273: the second variant outruns the first when using four cores, however, there seems to be saturation in that 8 cores are not doing better than 4 but slightly worse, why is this?

l.289-: I would delete these three last sentences

l.300: "$\sim$ 12 m metres"

l.303: -> Duncan (1954)

l.312: how did the winds change, such that the potential migration rate slowed down and not e.g. increased?

l.317: maybe add 'pollination' and 'this', i.e.: "overestimation of pollination distances, but this seems unlikely"

l.323: LAVESI-WIND?

l.324: maybe "evaluate with field-based data"?

l.336: from the perspective of a DGVM grid cell (typically > 0.5°) 5,000 x 5,000 m is still very small scale

l.337: LAVESI-WIND?

l.620: LAVESI-WIND?

l.621: "for each adult tree (potential pollen source) and for each seed source."

l.622 maybe write: "the shaded areas ... pollination probability for the labelled source ..." -> this is the probability for the one labelled source, or?

Fig 3.: I am surprised that 150-200m does not occur for pollination. Is this due to the perspective taken for the plot? I.e. when I understand it correctly the seeding is from the perspective away from a centre and the pollination is from the perspective of the centre? A short sentence on this would be great for a quick understanding. When looking solely at the figure it would also be nice to have some more information in the caption on what the figure shows, e.g. a short note if this is an average of 50 runs with the full LAVESI-WIND. Table 1: For a discussion of the re-parameterisation it would be nice if the original values of the parameters could be listed here, too. This would also help to identify the new parameters. Table 2: The different notations -15 and 345 are a bit irritating.

Supplementary

- l.8: note 5?

- "D.f." / "d.f." ? SS? F?

- Table S3: comma for thousand misplaced

- Table S3: why do the distance percentiles differ as compared to Table S2?

[Figure]

- Table S4: notation mixing initially confuses (-15 vs 345)

Additional References:

Huntley, B. , Barnard, P. , Altwegg, R. , Chambers, L. , Coetzee, B. W., Gibson, L. , Hockey, P. A., Hole, D. G., Midgley, G. F., Underhill, L. G. and Willis, S. G. (2010), Beyond bioclimatic envelopes: dynamic species' range and abundance modelling in the context of climatic change. Ecography, 33: 621-626. doi:10.1111/j.1600-0587.2009.06023.x

Lee, E. (2011). Impacts of Meteorology-Driven Seed Dispersal on Plant Migration: Implications for Future Vegetation Structure under Changing Climates. PhD thesis, Massachusetts Institute of Technology, Cambridge, Massachusetts.

Nabel, J.E.M.S. (2015). Upscaling with the dynamic two-layer classification concept (D2C): TreeMig-2L, an efficient implementation of the forest-landscape model TreeMig. Geosci. Model Dev., 8: 3563-3577. https://doi.org/10.5194/gmd-8-3563-2015

Snell, R. S., Huth, A. , Nabel, J. E., Bocedi, G. , Travis, J. M., Gravel, D. , Bugmann, H. , Gutiérrez, A. G., Hickler, T. , Higgins, S. I., Reineking, B. , Scherstjanoi, M. , Zurbriggen, N. and Lischke, H. (2014). Using dynamic vegetation models to simulate plant range shifts. Ecography, 37: 1184-1197. doi:10.1111/ecog.00580

Svenning, J. , Gravel, D. , Holt, R. D., Schurr, F. M., Thuiller, W. , Münkemüller, T. , Schiffers, K. H., Dullinger, S. , Edwards, T. C., Hickler, T. , Higgins, S. I., Nabel, J. E., Pagel, J. and Normand, S. (2014), The influence of interspecific interactions on species range expansion rates. Ecography, 37: 1198-1209. doi:10.1111/j.1600-0587.2013.00574.x

---

## Author Comment (AC1) · 30 Jun 2018

**Dear referee #1,**

**Thank you for positive review and helpful comments. We addressed each of them and separately responded in the following list:**

1. *"Sometimes more explanations would be very helpful, in particular concerning the pollination module."*

   **We followed the suggestion of the reviewer and extended the explanations of the newly implemented processes.**

2. *"In the introduction and discussion a lot of weight is put on migration processes and the inability of large-scale models to deal with it. In this context it seems at least a bit strange that the authors restrict their simulation to one plot of at maximum 200m\*200m. It would be more convincing to make clear that you started with a detailed small scale model which later could maybe inform larger scale models and in particular can enhance understanding of gene flow in forests."*

   **We extended the introduction with a statement about the intention of the model building as suggested.**

3. *"The question remains: how can this approach be adopted by models running on a larger scale? Can, for example, the detailed simulation of individual seeds and trees be upscaled? Does it make sense to use approaches as discussed in (Snell, 2014, Using dynamic vegetation models)? Please discuss."*

   **We extended the potential model applications paragraph with suggestions for further development and upscaling approaches with this model.**

4. *"As already pointed out using only a small area of 4 ha for studying migration processes seems weird. Also the circular boundary conditions might confound results of spatial processes because spatial gradients disappear. (Maybe I got that wrong, but then it indicates that a better explanation is needed!)"*

   **We used only a small area for the verification of the implemented processes: for the newly implemented sensitivity analyses we used 100 x 1,000 m transects with plots tied together on the meridional borders securing a south-to-north migration.**

5. *"It also seems to me that ignoring the size or age of father trees, and by that the relative amount they contribute to the pollen cloud, can affect the results."*

   **We did not explicitly include any allometric relationships of pollen production and tree performance in this version of the model because of sparse reference data, but it would be a further development. It is partly covered through the tree top height in the calculation of the pollination probability. See also our comment to  RC1#24. We added an explanation in the discussion.**

6. *"A lot would be clarified by more detailed captions of the figures."*

   **We added more descriptions to the figure captions.**

[revised manuscript text omitted]

---

## Author Comment (AC2) · 30 Jun 2018

**Dear referee #1,**

**We address the specific comments and clarify formulations:**

7. *"Throughout the text: avoid the term "pollination of seeds". Biologically the ovules are pollinated, and this yields the seeds. Or at least clarify that you mean the ovules in this context."*

   **We clarified here and throughout the manuscript, the process of pollination of ovules rather than seeds as suggested.**

8. *"L36: There are forest models for all scales, not only for stands. Omit "stand"."*

   **We adapted the sentence as suggested.**

9. *"L37: Between forest stand models and DGVMs, there are forest landscape models, which incorporate seed (whereas not pollen) dispersal (c.f. , Snell et al, 2014, Using dynamic vegetation models; Shifley et al., 2017) and have been applied also to the arctic (e.g. TreeMig, Epstein et al, 2007)."*

   **We included this model type in the enumeration.**

10. *"L50: what do you mean by "at the molecular level"? The genes? The coded proteins? Or the resulting traits?"*

    **We mean the traits of individuals, on which selection can act, and adapted the sentence for clarification.**

11. *"L50: Why does high gene flow constrain local adaptation? I would assume that it enhances it. Omit "high", or exchange "constrain" by "influence"."*

    **We edited the sentence to clarify the consequences of high gene flow for local populations.**

12. *"L51: I do not understand how local adaptation prevents negative consequences (which ones?) due to founder effects. Clarify"*

    **We clarified the consequences, see response RC1#11 for details.**

13. *"L63: commonly -> sometimes. E.g. concerning DGVMs, to my knowledge up to now Rebecca Snell's LPJ-GUESS version is the only one with something a bit similar to dispersal. Forest Landscape models of course contain dispersal, but believe it or not, it is very often switched off!"*

    **We exchanged the statement as suggested by the reviewer.**

14. *"L101: strict = absorbing?"*

    **Yes, we changed it.**

15. *"L108: It would biologically be more logical to start with pollination and describe seed dispersal afterwards"*

    **We exchanged the process implementation as suggested.**

16. *"L111: the formula is hard to read, one has to guess: (1/2)*distanceratio*(rand)**1.5 ???"*

    **We inserted an additional multiplicator sign for clear reading of the equation.**

17. *"L116: height of the tree top?"*

**We clarified the tree height by changing the expression to "tree top" as suggested.**

18. *"L120: Unclear: Probably you mean: wind speed and direction, which were randomly selected from a set of observations"*

    **We changed the sentence structure as suggested by the reviewer.**

19. *"L121: Start with a general conceptual description of the pollination and refer to fig. 1."*

    **We moved parts of the implemented pollination from the introduction to here and refer to the conceptual sketch in figure 1.**

20. *"L122: Is the pollination simulated really for each individual ovule? Idea for an upscaling: Would it be possible to do that more in a "share" approach per tree (assuming that all ovules are of the same genotype): Of N seeds of the tree, the share p1 are pollinated with pollen from father 1, p2 from father 2,..."*

    **Our description of the determination of pollen donors was not clearly stated: it was implemented in a "share" approach per tree as the reviewer suggests. We extended the description for clarification.**

21. *"L124: now you confuse me: tree or seed?"*

    **We clarified the statement.**

22. *"L131: make the reference to fig. 1 more prominent, I overread it several times, and found fig. 1 just by chance."*

    **Following the suggestion of the reviewer, we added a reference to the beginning of the paragraph.**

23. *"L138: "pollen distribution", isn't it rather the overall pollination probability of a seed?"*

    **We changed according to the suggestion the introduction of the equation.**

24. *"L140: Please make explicit if and how the pollen amount comes into play and how it depends on tree size or age or weather. How is a father tree defined by the variables above? How does its distance to the mother tree come into play? I see it in fig. 1, but not in the formulae"*

    **For simplification, we have not yet included further allometric relationships into the functions for pollination calculation, but this can easily be achieved. See further details in RC1#5. The distance between mother and father tree is used in the functions, variable $r$.**

25. *"L160: runaway line"*

    **We deleted the line.**

26. *"L187: What is the difference between the two parallelisations A and B?"*

    **We added a description of the two different ways of parralellisation of a STL-list-container.**

27. *"L189-205: I'm confused. What is the difference between the implemented dispersal function and the full model? Isn't the dispersal function implemented into the full model? So, where can differences come from? Why do you need this verification? Please clarify."*

    **We developed the functions in R and implemented them subsequent in C++. We intended to verify the implementation and correct computation of these and give a general overview about the resulting dispersal with our verification.**

28. *"L239: Why do you undergo the effort to calculate the wind-direction-dependent pollen transport, to then take father trees randomly from any direction???? Why not using the father trees in the different directions? Or did I misunderstand that? Do you really sample the father trees with uniform probability?"*

   **We edited the text to solve the potential misunderstanding of the calculation of the pollination process here and in the method section.**

29. *"L260: Why do you mention the times? Because the RAM adds up over time, or because the numbers of trees and seeds increase over time?"*

   **The list-container require RAM depending on the actual tree and seed elements stored in them.**

30. *"L284: and forest landscape models, in particular individual based ones like Iland (Seidl et al, 2012)"*

   **We added the additional model type of landscape models and the suggested citation.**

31. *"L312: Unclear. What would slow down the recent potential migration rate? And how?"*

   **We edited this and the preceding sentence for clarification.**

32. *"L358: This would require simulations on very large areas or long transects, for which the current approach is (still) too slow. Some simplification/upscaling would be required."*

   **We are currently working on a simplified model, which is informed by simulations with LAVESI-WIND but especially designed for millennial-scale continental simulations. According to the suggestion of the reviewer, we extended the sentence.**

33. *"L365-370: This is the real step forward, I would even emphasize it more!"*

   **We re-ordered the potential model application paragraph and placed this possible application at the end.**

34. *"Fig. 3: Please extend the caption, explaining in much more detail what was recorded. In particular the directions need to be explained. I guess, for wind and seeds, the direction means in which direction they go, in contrast for pollination the direction refers to where the father trees are?"*

   **We give more details in the caption for a better understanding. See also RC2#78.**

[revised manuscript text omitted]

---

## Author Comment (AC3) · 30 Jun 2018

**Dear referee #2,**

**Thank you for positive review and helpful comments. We addressed each of them and separately responded in the following list:**

1. *"New contributions are the insertion of wind speed and distance in the already existing seed dispersal kernel and the new pollination probability function. Beside this, however, it is only shown that the implementation of the functions work as intended and that two unspecified parallelisations lead to a computation time reduction. Else, the authors mainly write what potentially could be done with the model, but unfortunately miss to give sufficient instructions for how this could be done, nor do they show example simulations for the potential applications."*

   **We moved the verification to the supplement and included sensitivity analyses for both processes. In this new analyses, we used hypothetical transects and tracked for example the migration of the treeline border to evaluate the model's results to the change of the input parameters.**

2. *"I do not understand why the authors decided to put so much emphasis on describing the verification with all its sophisticated applied statistics. In my opinion the verification does not contribute to a knowledge gain and could be dealt with a few sentences, referring the main share to the supplementary. However, I do believe that a transparent description of the parameterisation and applied sensitivity analyses could be of large value for the community."*

   **This step was included to technically verify the implementation of wind-dependent seed dispersal and pollination. Following the recommendation of the reviewer we moved the verification to the supplement and included a sensitivity analyses for both newly implemented functions.**

3. *"I recommend a thorough revision of the manuscript targeting that in the end only those applications are emphasised in the abstract, introduction and conclusion that are sufficiently prepared in the methods and results and appropriately discussed."*

   **We edited the manuscript in many parts to include the new results of the sensitivity analyses.**

**Responses to recommendation for the different sections:**

4. "*(1) Abstract: In my opinion the abstract does currently not fit to the content of the paper. Reading it, I developed different expectations not addressed in the manuscript, such as model applications (at least exemplary) dealing with the advocated passing of genetic information and/or diversity and/or identifying important drivers of migration dynamics. Also global vegetation models are mentioned twice prominently in the abstract, e.g. last sentence: '... substantially help in unveiling the important drivers ...', however, unfortunately, I did not get an idea from the manuscript how this could be done and how global vegetation models can be improved. I recommend rewriting the abstract more closely describing the manuscript.*"

   **We extended the manuscript with sensitivity analyses for both processes and added the main results in the abstract. Our intention with this manuscript is to present the model improvements and to give in the abstract the key information about why we implemented the model extension.**

5. "*(2) Introduction: The authors refer to DGVMs and list processes not implemented in them. However, they do not mention why these processes are not included – e.g. issues of parameterisation (see e.g. Snell et al. 2014), particularly also due to the high variation in seed dispersal rates (and vectors) within the plant functional types currently used (see e.g. Huntley et al. 2010, Lee 2011, Snell et al. 2014, Svenning et al. 2014), issues of coarse spatial resolution and of computational complexity arising with inter-grid cell communication via seed dispersal or other spatially linked processes (e.g. Nabel 2015) – yet these issues that cause problems for larger scale models seem to even get visible in this manuscript describing a local single species model: when discussing the huge increase in computation cost and that the model is not properly parameterised for the suggested studies (indicated by sentences such as "Before applying this new model version, however, a proper parameterisation is necessary." "If this can be efficiently parameterised, the model could further be used to track genetic lineages in time."). I recommend mentioning these prominent problems of seed dispersal implementation in the introduction and relate to them in the discussion.*"

   **We refer to the parametrisation issue caused by the use of plant functional types in DGVMs as well as increased simulation complexity caused by inter-grid cell communication.**

6. "*(3) Methods: (a) According to 2.1 the presented simulations assume homogeneous forest stands with periodic boundary conditions. However, applications to study tree line dynamics, or migration in general, or the suggested "impact of migration processes on the genetics", require dispersal among inhomogeneous plots/gridcells and I would appreciate at least a small application showing how the model can be used for such conditions, showing e.g. what migration speed results from the model, which processes have largest influences etc.*"

   **We decided to follow the suggestion to include a sensitivity analysis for such processes. For this we simulated hypothetical 100 x 1,000 m transects on which seed dispersal was allowed on the meridional borders but not the latitudinal limits and tracked the colonisation to calculate migration rates. A further application to parameterise the model's functions and assess migration rates for north-central Siberia would be beyond the scope of this manuscript and was recently accepted for online discussion (https://doi.org/10.5194/bg-2018-267).**

7. *"(3) Methods: (b) "2.2.4 parameterisation to fit field data": even if the authors only visually compared the resulting number of trees, it might be helpful to have a table listing the number of trees for different parameter combinations (at least in the supplementary). What sensitivities were detected, which parameters were particularly important/ influential in tuning the number of trees? The authors state that the model parameters required a revision after implementing the extensions - what can we learn from this? It might be worth to discuss which parameters had to be revised and how, which are sensitive and influential - these things could be really helpful for the community."*

**We tuned manually the parameters and the resulting changed parameter values can be found in Table 1. In the Table we changed estimated to tuned in cases of fine tuning done for this manuscript. The regarding sensitivities can be found in Kruse et al. (2016) and also in the newly incorporated sensitivity analyses.**

8. *"(3) Methods: (c) Most of the parameters in Table 1 are stated as "estimated" but there at least needs to be a starting point! If the study aims to help parameterising DGVMs it would be helpful to at least know on what the parameters are originally based, even if its from another species etc."*

**We changed the status of parameters that were tuned here to "tuned", details about first guesses can be found in Kruse et al. (2016).**

9. *"(3) Methods: (d) 2.3.2: I find this section difficult to read and understand. If I am not mistaken the dispersal function was once implemented inside and once implemented outside the model (which I think is not explicitly stated somewhere?). I would move most of this and the according results section to the supplementary and only retain that the model was successfully verified."*

**We moved large parts of the verification as suggested, see RC2#2 for details.**

10. *"(3) Methods: (e) 2.3.3: The header of this section suggests an evaluation. The text of the section suggests that the simulation results will be quantitatively compared to observed values from literature. Yet I did not find such a comparison in the results section."*

**We have now included sensitivity analyses for both processes and moved the verification to the supplement. The resulting sensitivity values are discussed in the discussion section and we give further suggestions and comparisons to findings in the literature.**

11. *"(4) Results: (a) As stated above, I would move most of 3.1 to the supplementary (all comparison to 'direcly' estimated/calculated)."*

**We moved the verification of the implementation of the function to the supplement as the reviewer suggested.**

12. *"(4) Results: (b) I do not get the "fraction of significant differences" in Table 2"*

**We calculated the mean pollination distances based on the same set of mature individuals and compared the resulting distances to these calculations. In the Table we state the amount of significant different mean values for each of the directions.**

13. *"(4) Results: (c) The second parallelisation variant outruns the first when using four cores, however, there seems to be a saturation in that 8 cores are not doing better but even worse than 4, why is this? It would be interesting to see what happens with 16/more cores!"*

**We ran additional simulations as suggested. See comment RC2#65.**

14. "*(5) Discussion: (a) The authors start with stating that unlimited seedbeds cause high uncertainty in DGVM predictions. While I agree that the assumption of unlimited seedbeds contributes to the uncertainty in DGVM studies/projections I think that there are heaps of at least equally important uncertainty sources, like modelling of mortality, competition, canopy structures, etc. Furthermore, I do not see that the manuscript present a remedy for the uncertainty connected to seedbeds. In my opinion a min/max assumption with no and full dispersal currently gives a better uncertainty frame than difficult to parametrise dispersal kernels. If one of the main messages of the manuscript stays that LAVESI-WIND can be used to enhance DGVMs, e.g. by parameterising dispersal kernels, I strongly recommend to discuss the issue and difficulties of (PFT) parameterization (see e.g. Huntley et al. 2010, Lee 2011, Snell et al. 2014).*"

**We included further details to the suggested topics in the discussion and in the introduction. Corresponds with RC1#2.**

15. "*(5) Discussion: (b) LAVESI is a single species model parameterised for a certain Larix species. I wonder how not represented processes, such as competition with other species/understory, trait variability, or others, could influence migration processes, particularly in the currently tree-less tundra? Is this implicitly covered by the parameterisation? If it's implicitly covered, how might this change with climate change?*"

**Larches form single-species dominant forests in the areas the model was developed for and further competition with understory species (shrubs, grasses, etc.) were included implicitly. Future development of the model should be tested for the influence of such on the migration rate. We added a reference in the potential model application section.**

16. "*(5) Discussion: (c) Large parts of the model are realised as stochastic processes. How many repetitions are required to get a sound result, e.g. how is a targeted output variable such as migration speed influenced by the stochasticity?*"

**Even with the high stochasticity caused by many processes in the model, the variance is small when gathering the results of ten simulations. We randomly sampled from the 30 simulation repeats with a 50%-increased seed production factor the target variable of peak recruitment position. It can be seen that even with ten simulation repeats, the confidence interval is small and decreases further when using more repeats. Accordingly, we recommend to run a minimum of ten or better 30 simulation repeats to get sound results.**

[Figure]

**Figure review 1. Position of the peak recruitment position calculated from 2 to 30 simulation repeats relative to the value from 30 simulation repeats.**

17. *"(5) Discussion: (d) The last sentence of 4.2. stating that the model is not yet parameterisation to field data sounds as if it would only be a small step, but I assume that this is rather difficult and would like to read a bit more about how this can be done and what problems are anticipated."*

    **We explained that reference data for the northernmost arctic treeline areas are missing and thus a complete parameterisation could not be achieved.**

18. *"(5) Discussion: (e) The increase in computation costs associated with seed dispersal and the like have been mentioned in several places (e.g. Snell et al. 2014, Svenning et al. 2014, Nabel 2015) and are one of the reasons why seed dispersal is usually not accounted for on larger simulation extents. I think it would be worth to mention that this is a common finding when dealing with seed dispersal. Furthermore, the authors mention hundreds of CPUS here but did only conduct simulations with 1,4 and 8 CPUs (and with one of the parallelisation the avg sim time was worse with 8 than with 4 CPUs). In order to better support their statements and in order to assess the merit of the parallelisations another set of simulations with e.g. 16 or 32 CPUs would be helpful."*

    **We added accordingly the suggested citations in the text. Furthermore, we ran additional simulations to test whether the simulation time decreases further when using more CPUs. Furthermore, we found a wrong setting for the simulation of 8 cores and correcting this led to consistent results. See also comment RC2#65 for further details on this.**

19. *"(5) Discussion: (f) In my opinion section 4.4 is not well supported by the rest of the manuscript. I for example miss a link from simulations on the homogeneous 200 x 200 m plot with periodic boundaries to a study investigating migration rates. How are borders of simulation plots connected? How is the long distance dispersal dealt with and pollination which potentially reach over several plots? How does this increase the computational complexity, is such a simulation currently computationally feasible? Especially, when you a certain set of repetitions is required to account for the stochasticity?*

**For the simulations used in this manuscript, we stated the different model settings for boundary conditions in the methods section. The newly incorporated sensitivity analysis was run on a 100 wide and 1,000 m long hypothetical transect. The meridional borders are connected and seeds passing the borders are distributed further in the north or south depending on their dispersal direction. Pollen on the other hand is not distributed over the borders of simulated plots.**
**Whereas long-distance seed dispersal does not significantly increrase the simulation time, and the implementation of pollination over plot borders would be feasible, when traits are passed via pollen and seeds one must decide how the information is passed (e.g. sampling from all available trait space), which we partly included in the 'pollen influx' application.**

*Furthermore, how can pollen influx be studied when the amount of pollen is not simulated by the model?"*

**One simple approach we think about is to simulate forest stands similar to those around locations for which pollen influx rates are available from surface sediments or sediment cores and tune the pollination process (even by including an allocation factor) until the resulting rates match. Subsequently one can compare past influx rates to simulations with this parameterised model forced with climate data independent from the pollen record and evaluate the importance of other model parameters.**

[revised manuscript text omitted]

* only one observation, thus excluded from further analyses.

70

** Table S5.** – Comparison of pollination dispersal distances with Student's t-test of the mean p-values at a significance level of 0.01. Only cases with >10 pollination events of the five trees producing most seeds were considered.

| Cardinal direction of winds | Wind direction [°] | Mean p-value | Significance value (p) | Degrees of freedom | Statistic value (t) |
|---|---|---|---|---|---|
| North | 165-195 | 0.449 | <0.001 | 49 | 10.133 |
| | 135-165 | 0.277 | <0.001 | 38 | 5.629 |
| | 195-225 | 0.406 | <0.001 | 37 | 6.704 |
| South | 345-15 | 0.415 | <0.001 | 49 | 10.171 |
| | 15-45 | 0.302 | <0.001 | 40 | 5.734 |
| | 315-345 | 0.408 | <0.001 | 39 | 8.315 |

**Table S5.Table S6.** Test statistics for generalised nonparametric regression analyses (significance level: *** p<0.001).

| Simulation version | Model formula | Aikaike's Information Criterion (AIC) | Dispersion parameter for Gaussian family | Model term | D.f.Degrees of freedom | SSSum of squares | F-test statistic value | significance |
|---|---|---|---|---|---|---|---|---|
| **+POLLEN_PAR A** | t~Nt | 9641722 | 0.17 | Nt | 1 | 10160.4 | 59340.3 | *** |
| | | | | Residuals | 1076 | 184.2 | | |
| | t~Ns | 10421167 | 0.29 | Ns | 1 | 9093.8 | 31780.8 | *** |
| | | | | Residuals | 1076 | 307.9 | | |
| | t~Nt+Ns | 11671042 | 0.15 | Nt | 1 | 9879.2 | 65019.1 | *** |
| | | | | Ns | 1 | 7.3 | 48.2 | *** |
| | | | | Residuals | 1072 | 162.9 | | |
| | t~Nt+Ns+Nt:Ns | 1722964 | 0.14 | Nt | 1 | 10688.2 | 75655.2 | *** |
| | | | | Ns | 1 | 705.7 | 4995.0 | *** |
| | | | | Nt:Ns | 1 | 1843.4 | 13048.4 | *** |
| | | | | Residuals | 1071 | 151.3 | | |
| **+POLLEN** | t~Nt | 8841754 | 0.17 | Nt | 1 | 9730.8 | 56047.3 | *** |
| | | | | Residuals | 1077 | 187.0 | | |
| | t~Ns | 9751183 | 0.29 | Ns | 1 | 8857.5 | 30113.9 | *** |
| | | | | Residuals | 1077 | 316.8 | | |
| | t~Nt+Ns | 1183975 | 0.14 | Nt | 1 | 9472.3 | 66387.8 | *** |
| | | | | Ns | 1 | 9.0 | 63.3 | *** |
| | | | | Residuals | 1073 | 153.1 | | |
| | t~Nt+Ns+Nt:Ns | 1754884 | 0.13 | Nt | 1 | 10215.8 | 77934.1 | *** |
| | | | | Ns | 1 | 582.7 | 4445.4 | *** |
| | | | | Nt:Ns | 1 | 1647.4 | 12567.5 | *** |
| | | | | Residuals | 1072 | 140.5 | | |
| **+POLLEN_PAR B** | t~Nt | 7341596 | 0.15 | Nt | 1 | 9923.8 | 64828.7 | *** |
| | | | | Residuals | 1075 | 164.6 | | |
| | t~Ns | 8331045 | 0.25 | Ns | 1 | 8939.3 | 35072.6 | *** |
| | | | | Residuals | 1075 | 274.0 | | |
| | t~Nt+Ns | 1045833 | 0.13 | Nt | 1 | 9583.7 | 76495.4 | *** |
| | | | | Ns | 1 | 20.8 | 166.2 | *** |
| | | | | Residuals | 1071 | 134.2 | | |
| | t~Nt+Ns+Nt:Ns | 1596734 | 0.11 | Nt | 1 | 10405.4 | 91086.0 | *** |
| | | | | Ns | 1 | 681.4 | 5964.8 | *** |
| | | | | Nt:Ns | 1 | 1705.2 | 14927.1 | *** |
| | | | | Residuals | 1070 | 122.2 | | |
| **SEED** | t~Nt | -40340 | 0.05 | Nt | 1 | 1440.4 | 30730.1 | *** |
| | | | | Residuals | 1075 | 50.4 | | |
| | t~Ns | -375-233 | 0.06 | Ns | 1 | 1317.6 | 21817.7 | *** |
| | | | | Residuals | 1075 | 64.9 | | |
| | t~Nt+Ns | -233-375 | 0.04 | Nt | 1 | 1396.3 | 34099.6 | *** |
| | | | | Ns | 1 | 9.4 | 229.9 | *** |
| | | | | Residuals | 1071 | 43.9 | | |
| | t~Nt+Ns+Nt:Ns | 40-403 | 0.04 | Nt | 1 | 1512.3 | 37915.5 | *** |
| | | | | Ns | 1 | 37.9 | 950.9 | *** |
| | | | | Nt:Ns | 1 | 288.9 | 7243.1 | *** |
| | | | | Residuals | 1070 | 42.7 | | |

---

## Author Comment (AC4) · 30 Jun 2018

**Dear referee #2,**

**We address the "small comments, questions and remarks" below:**

20. "I.1 - title: Since LAVESI did already contain spatially explicit seed dispersal, maybe add wind driven "Implementing spatially explicit wind driven seed and pollen dispersal ...""

**We extended the title to clarify the scope of this manuscript.**

21. "I.23 - is the selected pollen donor still potential? maybe: "... select the pollen donor.""

**We adapted the sentence as suggested.**

22. "I.46 - "is mainly caused""

We improved the sentence to be clearer with the causation of overestimations of DGVMs.

23. "I.47 - "using a time lagged response function"? Snell 2014 uses a dispersal function for dispersal between grid cells and a logistic function to limit within grid cell dispersal""

The reviewer is right, we changed the sentence accordingly.

24. "I.53 - "this" refers to what? local adaptation?"

**We extended the content of the sentence and clarified the references.**

25. "I.59 - "this ensured the most realistic implementation" I would remove this part of the sentence or rephrase the paragraph, because of the previously mentioned dynamics at the treeline. I had to read the sentence several times before I realised that this refers to the homogeneous forest only."

We removed the unclear statement. Among other tested dispersal functions, the Gaussian with a fat tail led to stand structures best fitting to observations.

26. "I.63 - "are not coupled" – There is a phd thesis integrating wind dispersal in the DGVM CLM, see https://globalchange.mit.edu/sites/default/files/Lee\_PhD\_2011.pdf"

**We included the additional suggested reference, which is an important step.**

27. "I.64 - 'Among others, Student's 2Dt...' This sentence seems lost - delete it?"

**We removed the sentence as suggested.**

28. "I.69-77: move to methods or discussion"

We moved parts of the paragraph to the methods and also the discussion section.

29. "I.84: "Results of the validation" -> verification?"

**The regarding sentence was deleted.**

30. *"I.83-86: I would delete these sentences, particularly the second part informing that there is a discussion and conclusion section."*

**The sentence were restructured and now gives a better summary of the content of the manuscript.**

31. "I.93: I would use past here (i.e. "which were parameterised"), since you describe the parameterisation it in Kruse et al. (2016) and, if I understood it correctly, reparametrized it for this study?"

**We changed the use to past in the sentence.**

32. "I.95: What exactly are "homogeneous forest plots of 100 x 100 m" - 1. does a simulation runs on one plot or several plots? 2. Does homogeneous refer to the environmental conditions? What is the resolution and what is the extent of the simulation runs?"

Here, "homogeneous" refers to spatial homogeity, which means the climate is the across the whole plot. Wherever seeds are dispersed to within the borders of the plot, trees can grow. The plot size is variable, depending on the available computer sources, and a subgrid with 20-cm tiles is used for competition and further sensing of the environment of each tree. We edited the sentence for clarification.

33. "I.95: "seed dispersed beyond the plot borders" - thus out of the 100 x 100m plot?"

Yes.

34. "1.99: "... lag the hypothetical warming ..." – maybe "... densification and northwards migration might lag the applied hypothetical warming ...""

**We extended the sentence following the suggestion of the reviewer.**

35. "I.111: for better readability maybe insert a space or comma or other separator between distanceratio and rand"

We inserted an additional multiplicator sign for clear reading of the equation.

36. "I.113: Does this refer to (1) the sensitivity analysis in Kruse et al. 2016 or (2) a new analysis? If (1) maybe point again to the reference, if (2) can you show results of the sensitivity analysis in your supplementary?"

We based the estimates on our earlier sensitivity analysis documented in Kruse et al., 2016.

37. "I.116+117: do you have references for the 75%, for Vd and lambda? Is there any biological interpretation for lambda?"

The values of two-thirds of the tree top height was chosen here as a sophisticated estimate based on own observations and parameters Vd chosen for the species trait based on Matlack (1987). Lambda is a tuning parameter and here only used to bring the height from m to cm dimension. This parameter could further be used to set the dispersal distance differently for additional species.

38. "I.120: a reference to 2.2.5 would be helpful here"

We refer now to the wind observation data set description.

39. "I.124: "subsequently" to what?"

**We deleted the misleading word.**

40. "I.128: could you shortly write what C and m are? Why do you use m=1.25? Gregory (1961) seems to recommend different values for m for different conditions."

The parameters C and m are function parameters not further defined by Gregory (1961). We selected these values as a start here and fine tuning of these is planned in further studies.

41. "I.133: the symbols for the angle between trees and the actual wind direction are very difficult to visibly distinguish, because the bar is hardly noticeable. Maybe make the bar more visible, lower, or use another symbol for the wind direction"

**We changed the notation of theta for clarification.**

42. "I.143: "monthly climate series" - reading that the model has an annual temporal resolution I had to went back to the LAVESI description. I think it might be helpful to very briefly state which CRU variables are used to derive which annual driving variables"

**We refer now to the climate variables used and added a sentence to describe which driving variables are computed from these.**

43. "I.147: in this sentence field work is mentioned but it is neither explained what field work this was, nor a reference given. In I.156 I can infer that the reference probably is Wieczorek et al. (2017) and from this and other places I infer that the field work took place from 2011-2013. Maybe it would be easier to swap 2.2.3 and 2.2.4 and shortly properly introduce the field work."

**We changed the order of the paragraphs as suggested and added a brief introduction to the conducted fieldwork.**

44. "I.149: should probably be "-36°C""

**Yes, we corrected the temperature statement.**

45. "I.153: which field data? (see above). 100 x 100 areas -> is this the one 100x100m plot? How many repetitions did you simulate?"

**We improved the paragraph by additional explanation of the fieldwork.**

46. "I.155: I do not get which climate you use. Is this the climate described in 2.2.3 or not? If so: do not repeat, if not: why not?"

We deleted the additional reference to the field site and refer to the climate data, which is described in paragraph 2.2.4 below.

47. *"l.156: even if the number pf trees was only visually compared it would be nice to see some combinations of number of trees and parameters in a table"*

**We responded to this issue in RC2#7.**

48. "I.158: what is a good fit?"

**We clarified that we compared the stand densities for parameter tuning.**

49. "I.160: "Climate forcing data" – leftover?"

**We deleted the extra line.**

50. "I.171: do you refer with "plots" to the 50 repetitions of the one simulated plot?"

**Yes, we changed the text accordingly and refer to one plot rather than multiple plots.**

51. "I.182: from all pollen sources or from the selected donors?"

**We recorded only pollination events, not all potential pollen sources.**

52. "I.184: "implemented simulations" -> "conducted simulations""

We changed the word implemented to conducted as suggested by the reviewer.

53. "I.184-: repetition with 225"

For clarification, we merged the final sentence with some repetition into this.

54. "I.187: How do these two parallelisations differ?"

We added a description of the two different ways of parralellisation of a STL-list-container.

55. "I.203: "north between 345° and 15°" - how does this compare to the directions in Table 2?"

We changed the degree value of -15° that equates to 345° in Table 2 to the latter value to match the text.

56. "I.219: plural vs singular: "computation time increases""

We corrected the misuse of the plural of *time*.

57. "I.220: readability: "produced seeds""

We changed the order of words as suggested.

58. "I.225-: repetition with 184 (however without 2 CPUs)"

We deleted the sentence and checked given statements of used numbers of CPUs in the manuscript.

59. "I.239: uniform ? -> 2.2.2: "based on the probability density""

We deleted the misleading reference and clarified the selection of the father tree.

60. "I.242: "the higher values" - which?"

We clarified the connection of this and the preceding sentence.

61. "I.242-: I do not get: "4-6% of test show significant differences ... no evidence that the differences are significant"?"

We added the reference to the appropriate table 2 (now in supplement) with the stated fraction of significant differences.

62. "I.257: Maybe output storage is not in the Table S1?"

The output is written out on demand and not stored in the RAM to reduce the memory consumption especially for large simulated plots.

63. "I.269: maybe replace inferring by calculating/tracking"

We exchanged the word "inferring" to "including" for clarification.

64. "I.270: maybe replace estimating by calculating"

We exchanged the word as suggested.

65. *"I.273: the second variant outruns the first when using four cores, however, there seems to be saturation in that 8 cores are not doing better than 4 but slightly worse, why is this?"*

We set the parameters for the run with eight cores wrongly and used again only four. However, the computation time may increase due to overheads introduced due to idle cores when the tree list is split and handed over to a higher number of cores. This effect can be seen when comparing the two different parallelisation approaches, of which the second performs better with less cores but did not improve further when using more cores as we could see here for first variant.

66. "I.289-: I would delete these three last sentences"

We deleted these sentences as suggested.

67. "I.300: "~ 12 m metres""

We meant the maximum amount of seeds reaching ~12 m.

68. "I.303: -> Duncan (1954)"

We added the missing bracket.

69. *"I.312: how did the winds change, such that the potential migration rate slowed down and not e.g. increased?"*

This is a general statement and for clarification we edited this and the following sentences.

70. "I.317: maybe add 'pollination' and 'this', i.e.: "overestimation of pollination distances, but this seems unlikely""

We followed the advice of the reviewer and edited the sentence accordingly.

71. "I.323: LAVESI-WIND?"

We refer in the text now to the current model acronym as suggested.

72. "I.324: maybe "evaluate with field-based data"?"

We exchanged the word as suggested.

73. "I.336: from the perspective of a DGVM grid cell (typically > 0.5°) 5,000 x 5,000 m is still very small scale"

In this paper we present the implementation and first tests of wind-dependent seed dispersal and pollination in the individual-based spatially explicit model. A trial run needed 40 hours for 10,000 years of a dense forest plot simulation, showing the general applicability even for millennial simulations in time. This model did was not intended to be used for continuous continental scale simulations but more to use it as validation/paramterisation of interconnections and potential time lags between DGVM grids, however with further improvements and using large computer clusters this might be possible. We stated this in the end.

74. "1.337: LAVESI-WIND?"

We added the current model version acronym as suggested.

**75. "I.620: LAVESI-WIND?"**

**The version acronym was added to the model's name.**

76. "I.621: "for each adult tree (potential pollen source) and for each seed source.""

**We corrected the references to correspond with the figure legend.**

77. "I.622 maybe write: "the shaded areas ... pollination probability for the labelled source ..." -> this is the probability for the one labelled source, or?"

**We clarified the seed source reference in the caption.**

78. "Fig 3.: I am surprised that 150-200m does not occur for pollination. Is this due to the perspective taken for the plot? I.e. when I understand it correctly the seeding is from the perspective away from a centre and the pollination is from the perspective of the centre? A short sentence on this would be great for a quick understanding. When looking solely at the figure it would also be nice to have some more information in the caption on what the figure shows, e.g. a short note if this is an average of 50 runs with the full LAVESI-WIND.

The maximum pollination distance in the simulation setup is 155.6 m from one corner of the full 200 x 200 m plot to the most distant corner of the centre 20 x 20 m plot. We give more details in the caption for a better understanding.

Table 1: For a discussion of the re-parameterisation it would be nice if the original values of the parameters could be listed here, too. This would also help to identify the new parameters.

We decided to keep the table as it is and give here only the adapted values.

Table 2: The different notations -15 and 345 are a bit irritating."

We improved the notation. See comment RC2#83.

Further comments in the supplementary material:

79. "I.8: note 5?"

the structure Evaluation contains for each year one element. Footnote 5 was accidentally deleted.

80. ""D.f." / "d.f." ? SS? F?"

We give the full expressions in the table headers.

81. "Table S3: comma for thousand misplaced"

**We corrected the misplaced comma.**

82. "Table S3: why do the distance percentiles differ as compared to Table S2?"

The distance percentiles differ because here they are calculated for each direction separately and in the other table for both direction at the same time.

83. "Table S4: notation mixing initially confuses (-15 vs 345)"

We changed -15° to 345° for readability.

[revised manuscript text omitted]

---

## Referee Report (RR1)

**Review Kruse et al. : Implementing spatially explicit wind-driven seed and pollen dispersal in the individual-based larch simulation model: LAVESI-WIND 1.0**

My comments have been addressed. Some of the changes result in inconsistencies in the formulation. I've tried to mention them,  but the authors should check their text again.

My remaining minor comments

44: In Epstein 2007 the FLM TreeMig has been applied on a 2200 km transect from boreal to arctic conditions in Central Siberia, not exactly a small area. Forest landscape models are indeed what you seem to aim for in your outlook part in the discussion. So I would rather stress the novelty of including wind and pollen into such models.

78-86:   Mention the importance of pollen modelling, and smooth the transition to the follow-up sentence. ("Besides---traits")

130: is randomly determined according to this probability

147:  pollination probability, omit mathematical form

148: Just to make sure: p is the probability that a pollen donor standing in distance r to the mother tree is the father? If so, write probability p in line 129. Or is this probability normalized by the sum of the probabilities of all father trees?

211: does climate only influence tree growth or also survival and establishment (which are much more important)?

264: parametrisation

309: hand over? Unclear. I guess you distribute different tree individuals to different CPUs?

405-407: How can the simulation of a single species help to overcome the difficulties introduced by lumping several species into one artificial PFT??? Either explain or omit.

413-415: But how? You would still have the problem of intra-grid migration and the discretization error by the large grid cells. Have a look on the newest migration upscaling modelling approaches, which keep a fine resolution for the dispersal and simulate the local dynamics only in selected cells, e.g. Nabel 2015, and in particular Lehsten et al. 2018, GMD, *Simulating  migration in dynamic vegetation models efficiently with LPJ-GM*,

---

## Editor Decision (ED1)

Dear Dr. Stefan Kruse,

Please address the following items. This would be the last step for accepting your manuscript for publication.

**Item 1**

**A comment of the Reviewer #1:**
"405-407: How can the simulation of a single species help to overcome the difficulties introduced by lumping several species into one artificial PFT??? Either explain or omit."

**Your response:**
This statement holds (so far) only for the Siberian treeline ecotone with single-species dominated forests. Thus, we deleted the part of the sentence. However, we started to introduce several tree species of the Siberian boreal forests into the model and with this the model can be applied at a larger scale covering not only the latitudinal treeline ecotone.

**The corrected sentence (Line308~ in the rrevised manuscript):**
With this, it bears great potential to evaluate whether the difficulties caused e.g. by the plant functional type grouping many species with a variety of traits together as used in DGVMs (e.g. Lee 2011, Snell et al. 2014).

**My comment:**
I am not satisfied with your response here. Your model simulates a forest, where larch trees monopolize, and hence you can just avoid complicated works to classify varieties of woody species into smaller number of PFTs. There are no reason to expect that the model bears potential to evaluate difficulties caused by PFT classifications.

**Item 2**

**A comment of the Reviewer #2**
What is the exact transect for the sensitivity study - are these 10 100x100m plots with homogeneous climate (on each/all plots) or is this one 1000? It's a north-south transect, correct? In the review response the authors state that "seed dispersal was allowed on the meridional borders but not the latitudinal limits". I think this information is still missing in the current manuscript.

**Your response:**

The transect consists of only one plot. For these simulations we allow only establishment in the southernmost part of it in the beginning during stabilisation. In the following years, trees can establish in the remaining area northwards and the colonisation of the empty area can be observed.

We added the regarding information on the newly introduced mode of boundary conditions for seed dispersal only along meridional borders here and before in chapter 2.1. see also our answer to comment R2 [105].

**The corrected sentence (Line104~111 in the revised manuscript):**

The absorbing boundary condition had to be revised to allow the simulation of larger areas. Hence, we introduce a new mode of periodic boundary conditions that allows seeds leaving the simulated area to reenter on the opposite side, so that the borders of a simulation plot are connected along all borders. This mimics a tree stand within a homogeneous forest, similar to forest gap models (e.g. Brazhnik and Shugart, 2016; Pacala et al., 1996; Pacala and Deutschman, 1995; Zhang et al., 2011) and we used it in the simulations used for verification and paramterisation for this manuscript. A second mode was implemented for simulations of hypothetical north-south transects, which were used 110 in the sensitivity analyses, allowing seed dispersal only on the meridional borders but not the latitudinal limits.

**My comment:**

For making the description easily understandable, how about stating sizes of the periodic grid (100m×100m) and the transect (100m×1000m) here? Also, delete "110" in the last sentence.

---

## Author Response (AR2)

Dear Dr. Hisashi Sato,

Thank you for your helpful comments and we like to acknowledge the valuable suggestions and comments of both reviewers. We addressed each of them as well as the reviewer's comments and separately responded to them in the following list.

With kind regards on behalf of the authors,

Stefan Kruse

**List of responses to the comments and suggestions of the topical editor and the reviewers.**

**Comments of the topical editor Hisashi Sato:**

*"(1) Line 202 "lower most 100m wide"*
*Please add "x 100m length"."*

**We added the suggested clarification to the explanation of the area used in the stabilisation period for the sensitivity analyses.**

*"(2) Lines 313~314*
*I could not understand how you averaged the values in the table 3 to obtain these values (0.3 and 1.1)."*

**We referred to the mean absolute sensitivity values when parameters where changed by 5% of its reference value. However, we changed this here to include both levels of parameter change (5 and 50%). Additionally, mean absolute sensitivity values can now be found in table 3 as requested in R2: [Table 3].**

*"(3) Table 3*
*Two values in the table lack digits after the decimal point."*

**We checked all values and added missing digits.**

**Comments of the reviewer #1:**

*"44: In Epstein 2007 the FLM TreeMig has been applied on a 2200 km transect from boreal to arctic conditions in Central Siberia, not exactly a small area. Forest landscape models are indeed what you seem to aim for in your outlook part in the discussion. So I would rather stress the novelty of including wind and pollen into such models."*

**Following the comment of the reviewer, we stressed that state-of-the-art models lack wind-driven seed and especially pollen dispersal at the end of the sentence. Furthermore, we clarified that this is missing in Epstein's forest landscape model TreeMig.**

*"78-86: Mention the importance of pollen modelling, and smooth the transition to the follow-up sentence. ("Besides---traits")"*

**We restructured the beginning of the paragraph, which was also suggested by the second reviewer.**

*"130: is randomly determined according to this probability"*

**We extended the sentence for clarification.**

*"147: pollination probability, omit mathematical form"*

**We corrected the spelling of the first and deleted the second statement.**

*"148: Just to make sure: p is the probability that a pollen donor standing in distance r to the mother tree is the father? If so, write probability p in line 129. Or is this probability normalized by the sum of the probabilities of all father trees?"*

**We clarified the references to the probabilities in the regarding sentences.**

*"211: does climate only influence tree growth or also survival and establishment (which are much more important)?"*

**Several processes are depending directly (or indirectly) of the forcing climate. Here we list the influence on the main modules of the model LAVESI and refer for details to the publication with the model development (Kruse et al., 2016).**

*"264: parametrization"*

**We decided to follow Oxford spelling with the -s- variants of the words as it is requested by the 'manuscript preparation guidelines for authors' of GMD. In consequence, we did not change the spelling of the word here and in the manuscript.**

*"309: hand over? Unclear. I guess you distribute different tree individuals to different CPUs?"*

**Unfortunately, handing over individual tree individuals to functions would cause serious overheads because elements in the STL-container list are not directly accessible and one would have to iterate through the elements ahead until reaching the element of interest. The current structure of the model's source code does not allow this functionality. Nevertheless, we tried to clarify our parallelisation approach by using "compute" rather than "hand over" in the sentence.**

*"405-407: How can the simulation of a single species help to overcome the difficulties introduced by lumping several species into one artificial PFT??? Either explain or omit."*

**This statement holds (so far) only for the Siberian treeline ecotone with single-species dominated forests. Thus, we deleted the part of the sentence. However, we started to introduce several tree species of the Siberian boreal forests into the model and with this the model can be applied at a larger scale covering not only the latitudinal treeline ecotone.**

*"413-415: But how? You would still have the problem of intra-grid migration and the discretization error by the large grid cells. Have a look on the newest migration upscaling modelling approaches, which keep a fine resolution for the dispersal and simulate the local dynamics only in selected cells, e.g. Nabel 2015, and in particular Lehsten et al. 2018, GMD, Simulating migration in dynamic vegetation models efficiently with LPJ-GM"*

**We gave an example how to achieve the parameterisation and referred to the important publication of Lehsten et al., which is currently under discussions in GMD.**
**The upscaling approach of simplifying the computation time by connecting similar plots within a grid as done by Nabel (2015) seems very promising to reduce the computational demand of the model. This would be an potential hint in further model development for migration studies and population dynamics, however, to track the individuals from seeds to mature trees to assess the full genealogy through time this approach would unfortunately not be applicable. We referred to this interesting approach below in the discussions.**

**Comments of reviewer #2 Julia Nabel, general comments/suggestions:**

*"(1) What is the exact transect for the sensitivity study – are these 10 100x100m plots with homogeneous climate (on each/all plots) or is this one 1000? It's a north-south transect, correct? In the review response the authors state that "seed dispersal was allowed on the meridional borders but not the latitudinal limits". I think this information is still missing in the current manuscript."*

**The transect consists of only one plot. For these simulations we allow only establishment in the southernmost part of it in the beginning during stabilisation. In the following years, trees can establish in the remaining area northwards and the colonisation of the empty area can be observed.**
**We added the regarding information on the newly introduced mode of boundary conditions for seed dispersal only along meridional borders here and before in chapter 2.1. see also our answer to comment R2 [105].**

*"(2) In the review response the authors state that pollination only happens within a plot, i.e. if a transect is really made up of 10 100x100m plots pollination would not happen throughout the whole simulation area? I guess this is due to computational expenses? I would appreciate if the authors could mention this in 2.2.1 and add a few sentences on the limitations of this assumption for the potential model applications in 4.4."*

**The used transect for the simulation for this manuscript is a continuous area or plot and not made up of single simulated plots that are bound together. Therefore for the simulation of pollination all individuals on a transect are taken into account for, which strongly increases the computational effort. We stressed this and tried already for this manuscript to parallelise the source code in those parts in Chapter 4.3.2.**

*"(3) From current formula (2) – the authors seem to not have reordered the eq numbers – and the surrounding text I understand that Ht is the releasing tree top in m. However, in Table 2 it is listed as the factor 0.75. Furthermore, in the review response the authors state that lambda is a conversion factor from m to cm. Why is this conversion necessary? Aren't all the parameters and the result in m? Please clarify and also add the information about lambda to the manuscript."*

**We use in the program code centimeters for the variable 'seed releasing tree height' Ht so that we need to convert this for further usage in the dispersal. However, the unit conversion factor is misleading and for clarification, we deleted lambda in the equation and remained the statement of "Ht in m".**
**Furthermore, we added the correct abbreviations for the newly introduced factors for three parameters in Table 2.**

*"(4) Section 2.3 could benefit from a bit reordering, since information required to understand the first paragraph is only given in the second paragraph, namely that the lowermost 100 x 100 area (plot?) is the initialisation area and that each setting has been repeated 30 times."*

**As suggested, we reordered this chapter.**

*"(5) As already mentioned in (2) I would strongly appreciate if the authors could expand section 4.4 a bit more to state the limitations of LAVESI-WIND with regards to the advertised model applications. E.g. the authors state that migration behaviour could be influenced by shrubs which respond faster to climate change, but LAVESI is a one species model and might probably currently not be able to represent inter-species competition."*

**We moved the discussion of the general limitations of model applications in the discussion to section 4.4. as suggested from the reviewers. Furthermore, we clarified that shrubs are not represented in the model yet.**

**Minor comments/suggestions:**

*"[l46] maybe add that they have higher computational expenses, e.g. "... effort for parameterisation, have higher computational expenses, and are therefor typically not applied over larger areas"*

**We changed the sentence according to the suggestion of the reviewer.**

*"[l47] Further problems: of DGVMs?!"*

**We refer here to examples with DGVMs and clarified the sentence.**

*"[l64] I would move "during the past decades" to the start of the sentence"*

**We edited the sentence as suggested.**

*"[l77] "this pollination function" has no clear reference anymore, since sentences before have been deleted -- Maybe restructure sentence, e.g. "The new pollination function tracks ... and furthermore allows ..."*

**We followed the suggestions of the reviewer and edited the sentence.**

*"[105] Could you explain the new "periodic boundary conditions" a bit more? Is this the dispersal "allowed on the meridional borders but not the latitudinal limits" that you mention in the review responses? (In the sentence you also state that you use this for all simulations in the manuscript?)."*

**We edited the corresponding paragraph and clarified (1) the newly introduced modes for boundary conditions and (2) for which simulations these were used for in this manuscript.**

*"[112] individually -> individual ?"*

**The regarding word was corrected.**

*"[123] EQ numbers not reordered."*

**We updated the numbers of the equations throughout the manuscript.**

*"[125] maybe add further symbols and reorder the sentence a bit : "... velocity V_d_pollen... and wind speed Vw, and Gregory's parameters C and m"*

**We added the requested symbols and restructured the sentence.**

*"[148] Please add information on lambda."*

**This factor is now not any more used, see more information in our answer for comment R2: (3)**

*"[150] Why again wind speed, and how?"*

**We do not use wind speed for the estimation of the direction a second time so we deleted the wrong statement.**

*"[189] eq (5): is S the average of the 30 repetitions?"*

**We first calculate the sensitivity value for each individual simulation repeat and test if the distribution of resulting sensitivity values is significantly different from zero. In the tables (3 and 4) mean values are presented, and we added a short statement that these are calculated means over 30 repeats.**

*"[201] 10 100 x 100 plots?"*

**No, we use one 100 x 1,000 m simulation area. Please see our answer to comment R2: (1) for details.**

*"[260] Thus, which are the most sensitive parameters?"*

**We added the most important parameters with the highest sensitivity values to the end of this paragraph.**

*"[266] The ->max<- sensitivities increases?"*

**Yes, we refer here to the maximum values and clarified this in the sentence.**

*"[280] Line 282/283 state that "trees " are the most important variable, why is it then "number of seeds and combination of ... trees&seeds"?"*

**We deleted the wrong statement, which belonged to an earlier version of the performance assessment of the model.**

*"[289] "four to eight ... using only four"? Maybe "one to four ... using eight"?"*

**We edited the sentence and restructured it for clarification.**

*"[298] responses of the"*

**We corrected the spelling.**

*"[299] What exactly could be overcome?"*

**Deleted, see our answer in comment R1: 405-407**

*"[302] Maybe move parts of this to section 4.4., since it deals with potential applications"*

**We followed the suggestion of the reviewer and moved the last sentences to section 4.4.**

*"[339] I do not understand this sentence – the authors do have the data on the wind direction, thus they would know if this would have limited the recent migration rate?"*

**We have the data for the period of observations, which is generally not available for periods approximately before 1950 AD. This statement is just a warning that the choice of the wind input to force simulations could have a strong impact on the simulation outcome, which we now state in the sentence.**

*"[365] I would move this last paragraph to 4.4, since its a limitation of the potential application"*

**We integrated this paragraph to section 4.4. as suggested by the reviewer.**

*"[390] factor of six? results state a factor of 2!"*

**Yes, we corrected the wrong statement.**

*"[Table 3] Could you please add the mean absolute sensitivity referred to in the text? Could you add the information on bold/italic values from Table 4 also here in Table 3?"*

**We added the required mean absolute values and information in Table 3.**

[revised manuscript text omitted]

---

## Author Response (AR3)

Dear Dr. Hisashi Sato,
Please find our responses to your comments in the following. For each item, your comment and our response are highlighted in bold font.

The revised manuscript version with tracked changes is attached below.

With regards on behalf of the authors,
Stefan Kruse

**Comments of the topical editor:**
**Item 1**

*"A comment of the Reviewer #1:*
*"405-407: How can the simulation of a single species help to overcome the difficulties introduced by lumping several species into one artificial PFT??? Either explain or omit."*
*Your response:*
*This statement holds (so far) only for the Siberian treeline ecotone with single-species dominated forests. Thus, we deleted the part of the sentence. However, we started to introduce several tree species of the Siberian boreal forests into the model and with this the model can be applied at a larger scale covering not only the latitudinal treeline ecotone.*
*The corrected sentence (Line308~ in the rrevised manuscript):*
*With this, it bears great potential to evaluate whether the difficulties caused e.g. by the plant functional type grouping many species with a variety of traits together as used in DGVMs (e.g. Lee 2011, Snell et al. 2014).*
***My comment:***
***I am not satisfied with your response here. Your model simulates a forest, where larch trees monopolize, and hence you can just avoid complicated works to classify varieties of woody species into smaller number of PFTs. There are no reason to expect that the model bears potential to evaluate difficulties caused by PFT classifications."***

**Our response:**
**We intended to point out advantages of this single tree-species ecosystem in regards to simulating ecosystem responses with a detailed individual-based model in comparison to those done with global PFT-based simulation models. However, we accept your critic and deleted this misleading sentence in the introduction of our discussion.**

**Item 2**

*"A comment of the Reviewer #2*
*What is the exact transect for the sensitivity study - are these 10 100x100m plots with homogeneous climate (on each/all plots) or is this one 1000? It's a north-south transect, correct? In the review response the authors state that "seed dispersal was allowed on the meridional borders but not the latitudinal limits". I think this information is still missing in the current manuscript.*
*Your response:*
*The transect consists of only one plot. For these simulations we allow only establishment in the southernmost part of it in the beginning during stabilisation. In the following years, trees can establish in the remaining area northwards and the colonisation of the empty area can be observed.*
*We added the regarding information on the newly introduced mode of boundary conditions for seed dispersal only along meridional borders here and before in chapter 2.1. see also our answer to comment R2 [105].*
*The corrected sentence (Line104~111 in the revised manuscript):*
*The absorbing boundary condition had to be revised to allow the simulation of larger areas. Hence, we introduce a new mode of periodic boundary conditions that allows seeds leaving the simulated area to reenter on the opposite side, so that the borders of a simulation plot are connected along all borders. This mimics a tree stand within a homogeneous forest, similar to*

*forest gap models (e.g. Brazhnik and Shugart, 2016; Pacala et al., 1996; Pacala and Deutschman, 1995; Zhang et al., 2011) and we used it in the simulations used for verification and paramterisation for this manuscript. A second mode was implemented for simulations of hypothetical north-south transects, which were used 110 in the sensitivity analyses, allowing seed dispersal only on the meridional borders but not the latitudinal limits.*
*My comment:*
*For making the description easily understandable, how about stating sizes of the periodic grid (100m×100m) and the transect (100m×1000m) here? Also, delete "110" in the last sentence."*

**Our response:**
**We follow your suggestions and added in the regarding sentences the sizes of the simulated areas used for this manuscript for clarification. Furthermore, we deleted the erroneous statement of "110" in the last sentence.**

[revised manuscript text omitted]